# HiPO: Hybrid Policy Optimization for Dynamic Reasoning in LLMs

## Abstract

Large Language Models (LLMs) increasingly rely on chain-of-thought (CoT) reasoning to improve accuracy on complex tasks. However, always generating lengthy reasoning traces is inefficient, leading to excessive token usage and higher inference costs. This paper introduces the Hybrid Policy Optimization (i.e., HiPO), a framework for adaptive reasoning control that enables LLMs to selectively decide when to engage in detailed reasoning (Think-on) and when to respond directly (Think-off). Specifically, HiPO combines a hybrid data pipeline—providing paired Think-on and Think-off responses—with a hybrid reinforcement learning reward system that balances accuracy and efficiency while avoiding over-reliance on detailed reasoning. Experiments across mathematics and coding benchmarks demonstrate that HiPO can substantially reduce token length while maintaining or improving accuracy. Finally, we hope HiPO can be a principled approach for efficient adaptive reasoning, advancing the deployment of reasoning-oriented LLMs in real-world, resource-sensitive settings.

## 1 Introduction

Large Language Models (LLMs) have achieved unprecedented success across diverse cognitive tasks, from code generation and mathematical reasoning to scientific problem-solving. A key driver of this progress is the integration of **Chain-of-Thought (CoT)** (Yao et al., 2023; Wei et al., 2023) reasoning—a paradigm where models decompose complex queries into sequential, interpretable steps to derive accurate outputs. These approaches enhance accuracy on challenging problems but also introduce a persistent drawback: **overthinking** (Kumar et al., 2025; Sui et al., 2025; Nayab et al., 2025). Even for trivial queries, models often generate unnecessarily long reasoning chains, leading to inflated token usage, higher latency, and reduced efficiency in interactive applications. This inefficiency creates a fundamental tension between reasoning quality and computational cost, raising the need for mechanisms that can adaptively regulate reasoning depth.

Recently, recent work has explored adaptive reasoning control to mitigate overthinking, and can be divided into two categories: (i) training-based adaptive reasoning, where reinforcement learning (RL) (Aggarwal & Welleck, 2025; Arora & Zanette, 2025; Hou et al., 2025; Luo et al., 2025; Shen et al., 2025; Team et al., 2025; Lou et al., 2025) or supervised fine-tuning (SFT) (Munkhdalai et al., 2024; Ma et al., 2025; Chen et al., 2025a; Kang et al., 2025) encourages concise reasoning through length penalties or conciseness rewards; (ii) external control, which constrains reasoning with handcrafted prompts or dynamic instructions (Xu et al., 2025; Renze & Guven, 2024; Chen et al., 2024; Munkhbat et al., 2025). While effective to some extent, these methods suffer from important limitations: coarse supervision signals, monotonic incentives that discourage deeper reasoning on difficult problems, and a lack of principled trade-offs between accuracy, latency, and token efficiency.

To address these challenges, we introduce **HiPO** (**Hy**brid **P**olicy **O**ptimization), a unified framework for adaptive reasoning in LLMs. HiPO is designed to enable models to decide when to "think" (i.e., **Think-on**)and when to skip reasoning (i.e., **Think-off**), thereby striking a balance between correctness and efficiency. Specifically, our approach builds on two key innovations: (1) Hybrid Data Construction Pipeline. As shown in Figure 1, we first collect the training data containing both Think-on and Think-off responses. Each query is automatically categorized based on its difficulty and response correctness. Then, a high-performance model (i.e., DeepSeek-V3 (Liu et al., 2024a)) is used to produce the explicit explanations to justify its reasoning-mode decisions. Finally, for each query,

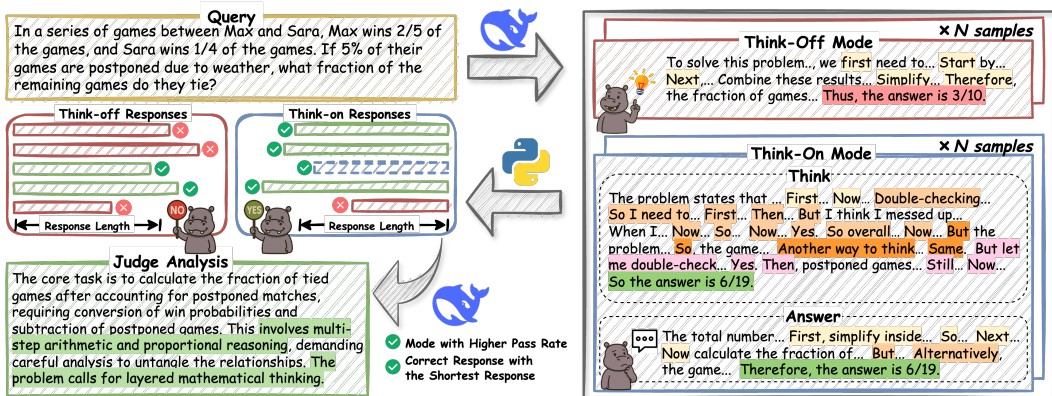

Figure 1: Framework of the hybrid data construction pipeline.

the final response based on the thinking mode and the corresponding explanation construct the hybrid output. (2) Hybrid Reinforcement Learning Reward System. We propose a hybrid reward design that balances Think-on and Think-off decisions. Specifically, a bias adjustment mechanism prevents the model from over-relying on verbose reasoning, while mode-aware advantage functions align reasoning-mode selection with actual performance gains. This ensures stable training and principled control over reasoning depth.

In summary, our contributions are threefold:

- We propose HiPO for adaptive LLM reasoning, which mainly includes the hybrid data construction and hybrid reinforcement learning.

- In the hybrid data construction pipeline, we produce logically rich Think-on and concise Think-off responses with the justification for the thinking mode. Then, for hybrid reinforcement learning, we introduce both the judge analysis and the response reward signal to enable principled control of reasoning depth.

- Experimental results on multiple datasets demonstrate that HiPO can consistently reduce redundant reasoning while improving or maintaining accuracy.

## 2 RELATED WORKS

**RL for LLM Reasoning.** Recent advances in reinforcement learning (RL) have significantly enhanced LLMs' complex reasoning capabilities, moving beyond supervised fine-tuning (SFT) limitations. State-of-the-art RL algorithms demonstrate superior performance in mathematical reasoning and multi-step problem solving: GRPO (Shao et al., 2024) stabilizes training through intra-group relative reward comparisons; GSPO (Zheng et al., 2025) defines sequence-level importance ratios and applies sequence-level clipping/rewarding/updates to improve efficiency and stabilize MoE training; VAPO (Yue et al., 2025) ensures reward consistency via value-aware optimization; PPO (Schulman et al., 2017) constrains policy updates through clipping mechanisms; and DPO (Rafailov et al., 2024) learns directly from human preferences without explicit reward modeling.

**Adaptive Reasoning.** Reasoning-oriented large language models—exemplified by Chain-of-Thought (CoT) (Yao et al., 2023; Wei et al., 2023) and R1-style (DeepSeek-AI et al., 2025) systems—have improved complex problem solving via explicit step-by-step reasoning and self-reflection but also suffer from "overthinking" (Kumar et al., 2025; Sui et al., 2025; Nayab et al., 2025), where simple queries trigger redundant chains that inflate compute, latency, and token usage, hindering interactive deployment. To address this, existing work focuses on: (i) Training-based adaptive reasoning: RL to conditionally trigger CoT, length penalties and conciseness rewards (Aggarwal & Welleck, 2025; Arora & Zanette, 2025; Hou et al., 2025; Luo et al., 2025; Shen et al., 2025; Team et al., 2025; Lou et al., 2025), and SFT (Munkhdalai et al., 2024; Ma et al., 2025; Chen et al., 2025a; Kang et al., 2025) to prefer shorter yet correct reasoning; (ii) External control : prompt or instruction designs that limit

steps or defer CoT (Xu et al., 2025; Renze & Guven, 2024; Chen et al., 2024; Munkhbat et al., 2025); (iii) Post-hoc Efficiency Optimization: pruning and restructuring chains after generation (Aytes et al., 2025; Xia et al., 2025; Liu et al., 2024b; Sun et al., 2024; Yang et al., 2025). Despite progress, these methods still face coarse supervision, limited adaptation to hard cases due to monotonic shortening, and a lack of principled trade-offs between quality, token cost, and latency.

## 3 METHOD

Our HiPO framework consists of two important components: (i) a hybrid data construction pipeline that generates training data with both Think-on and Think-off responses; (ii) a hybrid reinforcement learning reward system that combines mode-specific accuracy and global average performance, along with a bias-adjustment mechanism to prevent over-reliance on the Think-on mode.

### 3.1 HYBRID DATA CONSTRUCTION PIPELINE

This process begins with a novel data labeling system leveraging state-of-the-art LLMs to assess each query's difficulty and domain characteristics. Queries are then classified into Think-on and Think-off categories based on their intrinsic complexity and the availability of verifiable answers.

#### 3.1.1 DATA SOURCE

We construct a challenging corpus for code and mathematics by integrating diverse public and proprietary sources, as illustrated in Fig. 2, including AM-Thinking-v1-Distilled (Tian et al., 2025), II-Thought-RL (Internet, 2025), AceReason-Math (Chen et al., 2025b), and Skywork-OR1-RL-Data (He et al., 2025).

#### 3.1.2 DATA COLLECTION

To effectively enhance the performance of HiPO, we design a structured data construction pipeline aimed at exploring and guiding the model's preference between the Think-on and Think-off reasoning modes. Our training dataset is meticulously curated to be logically rich, cross-domain, and sufficiently challenging.

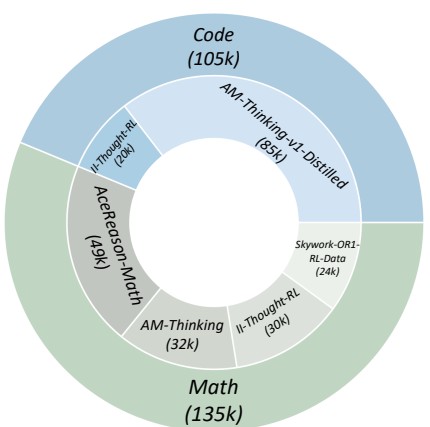

Figure 2: Statistics of Data Sources.

We adopt a multi-stage data generation process as shown in Figure 1. For each query, the pipeline samples $N$ responses under the Think-on mode and $N$ responses under the Think-off mode using a dedicated reasoning model. All responses are then verified for correctness, and the reasoning mode with the higher pass rate is selected as the preferred mode for that query. Let $p_{on}$ and $p_{off}$ denote the pass rates of the Think-on and Think-off modes, respectively. If the difference in pass rates satisfies $|p_{on} - p_{off}| < \delta$, where $\delta$ is a predefined threshold, the Think-off mode is selected. This tie-breaking strategy encourages the model to prefer more concise responses when deeper reasoning does not lead to a significant improvement in correctness. For the winning mode, the shortest correct response is retained as the final sample. To expose the model to diverse reasoning scenarios and encourage adaptive behavior, we randomly assign a mode to 1% of the queries, forcing the model to encounter diverse reasoning scenarios. This forces the model to engage with both reasoning styles in varying contexts, which is essential for learning when to switch modes dynamically during inference. Additionally, we incorporate an auxiliary explanation signal to enhance the model's mode alignment capabilities. For each query-response pair, we prompt DeepSeek-V3 (Liu et al., 2024a) to generate a justification explaining why the selected mode is appropriate. This explanation provides a valuable training signal for aligning mode decisions with the underlying reasoning complexity.

Table 1: Formatting templates (left) and special tokens with their descriptions (right).

| Think-on Mode | Think-off Mode |
|---|---|
| `<judge>`
`{judge_analysis}`
`</judge>`

`<think_on>`
`<think>`
`{thinking_content}`
`</think>`

`<answer>`
`{response}`
`</answer>` | `<judge>`
`{judge_analysis}`
`</judge>`

`<think_off>`
`<answer>`
`{response}`
`</answer>` |

| Special Token | Description |
|---|---|
| `<judge>` | Analyzes input query to determine whether reasoning is required. |
| `<think_on/off>` | Specifies whether reasoning should be activated ("on") or skipped ("off"). |
| `<think>` | Marks the beginning of reasoning in Think-on mode. |
| `<answer>` | Marks the beginning of the model's answer. |

### 3.1.3 DATA FORMAT

The training samples follow a unified structure encompassing justification and answer generation. As shown in Table 1, this design guides the model to decide when reasoning is needed and to generate answers consistent with it. The special tokens are detailed in Table 1, ensuring a clear separation between reasoning and final response for better alignment.

## 3.2 HYBRID RL REWARD SYSTEM

This section details the reinforcement learning process used to teach the model how to effectively balance Think-on and Think-off reasoning modes. The approach is built on a hybrid RL reward system that guides the model's optimization.

### 3.2.1 BASIC REWARD FORMULATION

Consider a group of $N$ sampled responses, for each response $i \in \{1, \ldots, N\}$, we denote its answer correctness by $\mathrm{ACC}_i \in \{0, 1\}$, its format correctness by $\mathrm{FORMAT}_i \in \{0, 1\}$, its basic reward by $r_i = \mathrm{ACC}_i + 0.2 \cdot \mathrm{FORMAT}_i \in \mathbb{R}$, and its reasoning mode by $M_i \in \{\mathrm{on}, \mathrm{off}\}$, where $M_i = \mathrm{on}$ indicates the Think-on mode and $M_i = \mathrm{off}$ indicates the Think-off mode.

### 3.2.2 BIAS ADJUSTMENT MECHANISM

A potential risk of the hybrid reward design is that the model may overfit to the more accurate Think-on mode, favoring deep reasoning even when it is unnecessary. This tendency can reduce response efficiency and hinder the intended flexibility in reasoning behavior. To mitigate this issue, we introduce a bias adjustment mechanism that dynamically regularizes the contribution of mode-specific accuracies.

Let $\mathrm{mean}(\mathbf{r}_{\mathrm{on}}) = \frac{1}{N_{\mathrm{on}}} \sum_{i:M_i=\mathrm{on}} r_i$ denote the average reward of responses generated under the Think-on mode, and let $\mathrm{mean}(\mathbf{r}_{\mathrm{off}})$ denote the corresponding average reward for the Think-off mode. Based on this, we define a bias term for the Think-off mode as a fraction of the Think-on average reward: $\mathrm{bias}_{\mathrm{off}} = \omega \cdot \mathrm{mean}(\mathbf{r}_{\mathrm{on}})$, where $\omega$ controls the ratio. The adjustment is applied only when the performance of the Think-off mode does not exceed that of the Think-on mode, but the difference between the two remains within the bias threshold. Formally, the adjustment mechanism is as follows:

$$\mathrm{mean}(\mathbf{r}_{\mathrm{off}}) = \begin{cases} \mathrm{mean}(\mathbf{r}_{\mathrm{off}}) + \mathrm{bias}_{\mathrm{off}}, & 0 \leq \mathrm{mean}(\mathbf{r}_{\mathrm{on}}) - \mathrm{mean}(\mathbf{r}_{\mathrm{off}}) \leq \mathrm{bias}_{\mathrm{off}}, \\ \mathrm{mean}(\mathbf{r}_{\mathrm{off}}), & \text{otherwise.} \end{cases} \tag{1}$$

This mechanism prevents the model from gaining an unfair advantage by overfitting to the more verbose but more accurate Think-on mode. Moreover, it ensures that the adjusted accuracies remain faithful to the true relative performance between reasoning modes, thereby improving training stability and preserving the intended balance between depth and efficiency.

### 3.2.3 SUPERVISION RL WITH HIPO

The final advantage function is formulated as a hybrid signal that integrates both judge analysis and model response. Each response $i$ receives two distinct scalar advantage, including *judge advantage* based on the quality of the mode justification, and *answer advantage* based on correctness and format.

The judge advantage $A_i^{\text{judge}}$ captures the broader decision-level utility of selecting a particular mode. The first term, $\text{mean}(\mathbf{r}_{M_i}) - \text{mean}(\mathbf{r})$, quantifies the global advantage of the chosen mode over the full group average, guiding the model toward choosing modes that lead to higher expected rewards. The second term, $\gamma \cdot (r_i - \text{mean}(\mathbf{r}))$, ensures that the justification content is also responsible for the quality of the response under that mode, thereby aligning the explanation with actual performance. The use of a global normalization factor $\text{std}(\mathbf{r})$ stabilizes the reward signal across groups. The judge advantage function for response $i$ is then given by:

$$A_i^{\text{judge}} = \begin{cases} \frac{(\text{mean}(\mathbf{r}_{\text{on}}) - \text{mean}(\mathbf{r})) + \gamma \cdot (r_i - \text{mean}(\mathbf{r}))}{\text{std}(\mathbf{r})}, & \text{if } M_i = \text{on}, \\ \frac{(\text{mean}(\mathbf{r}_{\text{off}}) - \text{mean}(\mathbf{r})) + \gamma \cdot (r_i - \text{mean}(\mathbf{r}))}{\text{std}(\mathbf{r})}, & \text{if } M_i = \text{off}. \end{cases} \quad (2)$$

In contrast to the judge advantage function, the advantage $A_i^{\text{answer}}$ is computed within the context of the selected reasoning mode. Since the mode $M_i$ has already been determined prior to response generation, it is natural to assess the response quality relative to other responses within the same mode. This local normalization using mode-specific mean and standard deviation focuses the learning signal on intra-mode variance, encouraging the model to improve response quality without conflating mode preference. For response $i$, the answer advantage is defined as:

$$A_i^{\text{answer}} = \begin{cases} \frac{r_i - \text{mean}(\mathbf{r}_{\text{on}})}{\text{std}(\mathbf{r}_{\text{on}})}, & \text{if } M_i = \text{on}, \\ \frac{r_i - \text{mean}(\mathbf{r}_{\text{off}})}{\text{std}(\mathbf{r}_{\text{off}})}, & \text{if } M_i = \text{off}. \end{cases} \quad (3)$$

To assign token-level reward for training with reinforcement learning, we define the final reward for each token $t$ in sample $i$ as follows:

$$A_{i,t} = \begin{cases} A_i^{\text{answer}}, & \text{if token } t \in \mathcal{T}^{\text{answer}}, \\ A_i^{\text{judge}}, & \text{if token } t \in \mathcal{T}^{\text{judge}}. \end{cases} \quad (4)$$

where $\mathcal{T}^{\text{judge}}$ and $\mathcal{T}^{\text{answer}}$ denote the token index sets corresponding to the judge segment and the answer segment, respectively, within each response.

Given a query $q$, HiPO generates a collection of candidate outputs $\{o_i\}_{i=1}^G$ from the old policy $\pi_{\theta_{\text{old}}}$. For each output $o_i$, let $\mathcal{T}_i$ denote the set of token positions in response $i$, i.e., $\mathcal{T}_i = \mathcal{T}^{\text{judge}} \cup \mathcal{T}^{\text{answer}}$. We define the per-token probability ratio as $\rho_{i,t} = \frac{\pi_\theta(y_{i,t} \mid h_{i,t})}{\pi_{\theta_{\text{old}}}(y_{i,t} \mid h_{i,t})}$, where $y_{i,t}$ is the $t$-th generated token in $o_i$ and $h_{i,t}$ is its conditioning context. The policy $\pi_\theta$ is optimized by maximizing the following token-level objective:

$$\mathcal{J}(\theta) = \mathbb{E}\Big[q \sim P(Q), \{o_i\}_{i=1}^G \sim \pi_{\theta_{\text{old}}}(\cdot \mid q)\Big]$$

$$\cdot \frac{1}{G} \sum_{i=1}^G \frac{1}{|\mathcal{T}_i|} \sum_{t \in \mathcal{T}_i} \Big( \min\big(\rho_{i,t} A_{i,t},\ \text{clip}(\rho_{i,t}, 1-\epsilon, 1+\epsilon)\, A_{i,t}\big) \quad (5)$$

$$- \beta\, \mathbb{D}_{KL}\big(\pi_\theta(\cdot \mid h_{i,t}) \,\|\, \pi_{\text{ref}}(\cdot \mid h_{i,t})\big)\Big).$$

Here $A_{i,t}$ is the token-level advantage defined in Eq. (4) via segment-wise assignment, and $\mathbb{D}_{KL}$ is the token-level KL between the current policy and the reference policy at context $h_{i,t}$.

## 3.3 TRAINING PARADIGM

Our HiPO framework adopts a two-stage training paradigm, consisting of a **cold-start** stage and a **RL** stage. In the code-start stage, the model is initialized with high-quality, hybrid training data that contains both Think-on and Think-off responses. This stage enables the model to acquire fundamental reasoning and answering capabilities, while establishing an initial balance between analytical reasoning and concise responses. In the RL stage, the model is further optimized using our hybrid reward system, which integrates mode-specific accuracy and global average performance. Together, these two stages ensure that HiPO achieves both strong factual accuracy and robust reasoning ability across diverse domains.

## 4 EXPERIMENTS

### 4.1 EXPERIMENTAL SETUP

**Implementation details.** Since the Qwen3 model can freely switch between inference modes, we chose it for our experiment. However, when the training data is insufficient, training the Qwen3 model can easily lead to a decline in performance on the test set (details can be found in the appendix A.2). To address this, we conducted Cold-Start tuning to stabilize its performance with relatively large datasets. For the Cold-Start stage, we use the "AM-Thinking-v1-Distilled", "AceReason-Math", "AM-Thinking", "II-Thought-RL(math)" dataset for training. The parameters are set as: maximum learning rate is 8e-5, minimum learning rate is 8e-6 and batch size is 512. For the RL stage, we use the "II-Thought-RL(code)", "Skywork-OR1-RL-Data" dataset for training. The parameters are set as: batch size = 16, maximum response length = 32k, $N = 16$, $\omega = 0.01$, and $\gamma = 0.3$.

**Baselines.** To demonstrate the effect of mitigating overthinking, we designed the following baselines for comparison. (1) **Cold-Start**: We perform Cold-Star on the model using the data construction method described in Section 3.1. (2) **Cold-Start (On)**: We apply the same Cold-Star procedure as in Section 3.1, but only include the data collected under the Think-on mode. (3) **Cold-Start (On) + GRPO**: We further train the **Cold-Start (On)** model using the GRPO algorithm. (4) **Cold-Start + GRPO**: We further train the **Cold-Start** model with the GRPO algorithm. (5) **HiPO**: We train the model following our HiPO. (6) **AdaptThink**: We reproduced the code provided in (Zhang et al., 2025). (7) **AutoThink**: We reproduced the code provided in (Tu et al., 2025).

**Evaluation benchmarks.** We conducted tests on AIME2024, AIME2025, HumanEval (Chen et al., 2021), LiveCodeBench V6 (Jain et al., 2024), MBPP (Austin et al., 2021), MATH-500 (Lightman et al., 2023), and GPQA-Diamond (Rein et al., 2023).

### 4.2 MAIN RESULTS

| Method | AIME2024 | | | AIME2025 | | | LiveCodeBench | | | HumanEval | | |
|---|---|---|---|---|---|---|---|---|---|---|---|---|
| | Acc↑ | Length↓ | Ratio_T↓ | Acc↑ | Length↓ | Ratio_T↓ | Acc↑ | Length↓ | Ratio_T↓ | Acc↑ | Length↓ | Ratio_T↓ |
| Cold-Start (on) | 80.8 | 21265 | 1.00 | 71.7 | 23791 | 1.00 | 56.2 | 19473 | 1.00 | 82.9 | 2662 | 1.00 |
| + GRPO | 82.5↑2.1% | 21045↓1.0% | 1.00−0.0% | 76.7↑7% | 22695↓4.6% | 1.00−0.0% | 57.3↑2.0% | 19067↓2.1% | 1.00−0.0% | 95.1↑14.7% | 3597↑35.1% | 1.00−0.0% |
| Cold-Start | 85.8↑6.2% | 18138↓14.7% | 1.00−0.0% | 76.7↑7.0% | 20613↓13.4% | 1.00−0.0% | 60.8↑8.2% | 18158↓6.8% | 0.91↓9.0% | 88.4↑6.6% | 2272↓14.6% | 0.54↓46.3% |
| + GRPO | 86.7↑7.2% | 17083↓19.7% | 1.00−0.0% | 79.17↑10.5% | 19869↓16.5% | 1.00−0.0% | 62.1↑10.6% | 18046↓7.3% | 0.93↓7.3% | 87.8↑5.9% | 2220↓16.6% | 0.59↓40.8% |
| AdaptThink | 83.3↑3.1% | 16598↓21.9% | 0.93↓7.0% | 74.2↑3.5% | 19993↓16.0% | 0.84↓16.0% | 57.1↑1.6% | 16162↓17.0% | 0.78↓28.0% | 85.4↑3.0% | 915↓65.6% | 0.16↓84.0% |
| AutoThink | 84.3↑3.5% | 17061↓19.8% | 0.95↓5.0% | 75.0↑4.6% | 18784↓21.0% | 0.88↓12.0% | 57.5↑2.3% | 15672↓19.5% | 0.80↓20.0% | 82.3↓0.7% | 1050↓60.6% | 0.18↓82.0% |
| HiPO | 87.5↑8.3% | 15107↓29.0% | 0.98↓1.7% | 82.5↑15.1% | 17655↓25.8% | 0.95↓5.0% | 63.0↑12.2% | 13558↓30.4% | 0.82↓18.5% | 90.2↑8.8% | 776↓70.9% | 0.12↓88.4% |

| Method | MATH-500 | | | GPQA-Diamond | | | MBPP | | | Average | | |
|---|---|---|---|---|---|---|---|---|---|---|---|---|
| | Acc↑ | Length↓ | Ratio_T↓ | Acc↑ | Length↓ | Ratio_T↓ | Acc↑ | Length↓ | Ratio_T↓ | Acc↑ | Length↓ | Ratio_T↓ |
| Cold-Start (on) | 92.0 | 6237 | 1.00 | 61.1 | 10832 | 1.00 | 72.0 | 4411 | 1.00 | 73.8 | 12667 | 1.00 |
| + GRPO | 93.2↑0.0% | 6256↑1.3% | 1.00−0.0% | 57.6↓5.8% | 10633↓1.8% | 1.00−0.0% | 71.8↓0.3% | 5103↑15.7% | 1.00−0.0% | 76.3↑3.3% | 12628↓0.3% | 1.00−0.0% |
| Cold-Start | 93.0↑1.1% | 5215↓16.4% | 0.65↓35.0% | 61.6↑0.8% | 11172↑3.1% | 0.95↓4.5% | 71.4↓0.8% | 3561↓19.3% | 0.42↓58.0% | 76.8↑4.1% | 11304↓10.8% | 0.78↓21.8% |
| + GRPO | 92.8↑0.9% | 5204↓16.6% | 0.68↓31.8% | 58.6↓4.1% | 10581↑2.3% | 0.98↓2.0% | 72.0−0.0% | 4341↑1.6% | 0.38↓62.0% | 77.0↑4.3% | 11049↓12.8% | 0.79↓20.6% |
| AdaptThink | 92.8↑0.9% | 4213↓32.5% | 0.55↓45.0% | 56.1↓8.2% | 10242↓5.4% | 0.91↓9.0% | 68.0↓5.6% | 4165↓5.6% | 0.33↓67.0% | 73.8−0.0% | 10327↓18.5% | 0.64↓36.0% |
| AutoThink | 92.8↑0.9% | 4261↓31.7% | 0.56↓44.0% | 58.1↓4.9% | 9898↓8.6% | 0.89↓11.0% | 70.0↓2.8% | 4958↓12.4% | 0.40↓60.0% | 74.3↑0.7% | 10240↓19.2% | 0.67↓33.0% |
| HiPO | 93.6↑1.7% | 4090↓34.4% | 0.54↓45.8% | 60.1↓1.7% | 9367↓13.5% | 0.92↓−8.1% | 72.2↑0.3% | 1338↓69.7% | 0.12↓88.0% | 78.4↑6.3% | 8842↓30.2% | 0.63↓36.5% |

Table 2: Based on Qwen3-8B, performance of different methods on multiple benchmarks. **Ratio_T** denotes the ratio of "Think-on" mode over the corresponding benchmark.

In Table 2, we observe that training the model solely on Think-on data leads the model to engage in reasoning for problems of any difficulty. We use this baseline as a typical example of "overthinking" for comparison. After applying GRPO to the Cold-Start (on) model, there is a significant improvement in accuracy, with an average accuracy increase of 3.1%. However, this does not reduce the token length and thinking rate of the model. On the contrary, to achieve higher accuracy, the token length output by the model on simpler datasets increases significantly. When training the model on a dataset containing both Think-on and Think-off data, the accuracy of the resulting Cold-Start model improves by 4.0% compared to the Cold-Start(on) model, while the token length and thinking rate decrease

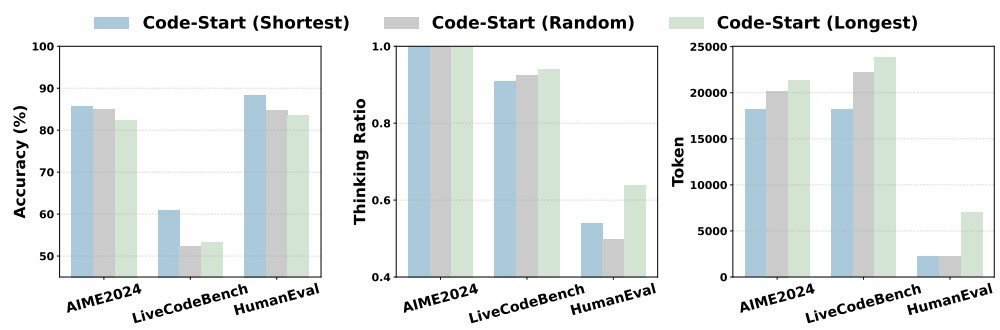

Figure 3: Performance of different response selection strategies.

| Method | AIME2024 | | | LiveCodeBench | | | HumanEval | | |
|---|---|---|---|---|---|---|---|---|---|
| | Acc | Length | Ratio$_T$ | Acc | Length | Ratio$_T$ | Acc | Length | Ratio$_T$ |
| HiPO | 87.50 | 15107 | 0.98 | 63.00 | 13558 | 0.82 | 90.2 | 776 | 0.12 |
| HiPO (w/o global adv) | 85.83$_{\downarrow1.9\%}$ | 18064$_{\uparrow19.6\%}$ | 1.00$_{\uparrow2.0\%}$ | 56.83$_{\downarrow9.8\%}$ | 14561$_{\uparrow7.4\%}$ | 0.86$_{\uparrow4.9\%}$ | 89.63$_{\downarrow4.9\%}$ | 1660$_{\uparrow114.9\%}$ | 0.27$_{\uparrow125.0\%}$ |
| HiPO (w/o local norm) | 85.00$_{\downarrow2.9\%}$ | 18268$_{\uparrow20.9\%}$ | 1.00$_{\uparrow2.0\%}$ | 58.37$_{\downarrow7.3\%}$ | 16029$_{\uparrow18.2\%}$ | 0.88$_{\uparrow7.3\%}$ | 89.63$_{\downarrow0.6\%}$ | 2052$_{\uparrow164.4\%}$ | 0.32$_{\uparrow166.7\%}$ |

Table 3: Performance of different design strategies on advantage functions.

by 10.8% and 22%, respectively. After applying the GRPO algorithm to the Cold-Start model, there is no significant change in performance. However, when applying our method HiPO to train the Cold-Start model, the accuracy improves by 6.2%, while the token length and thinking rate decrease dramatically by 30% and 39%, respectively. Moreover, experimental results show that our HiPO outperforms existing methods on both efficiency and accuracy.

## 4.3 ABLATION STUDY

**Effect of selecting the shortest response.** In the data construction pipeline, we select the shortest response (Cold-Start (Shortest)) as the final sample. To analyze the effect of this strategy, we additionally propose two variants called (**Cold-Start (Longest)** and **Cold-Start (Random)**) by selecting the longest responses and randomly selecting the responses, respectively. In Figure 3, Cold-Start (Shortest) shows an improvement in accuracy compared to both Cold-Start (Longest) and Cold-Start (Random), with a decrease in both the Thinking ratio and Token length. Therefore, we adopt this Cold-Start (Shortest) strategy for the Cold-Start stage.

**Effect of design strategies for $A_i^{\text{judge}}$ and $A_i^{\text{answer}}$.** In the reinforcement learning stage, first, we utilize the term $\text{mean}(\mathbf{r}_{M_i}) - \text{mean}(\mathbf{r})$ to quantify the **global advantage** of the chosen mode over the full group average. Second, the **local normalization** based on the mode-specific mean and standard deviation is used for $A_i^{\text{answer}}$. To demonstrate the effect of these strategies, as shown in Table 3, we design two variants (i.e., HiPO (w/o global adv) and HiPO (w/o local norm)). For HiPO (w/o global adv), we directly remove the global advantage for $A_i^{\text{judge}}$. For HiPO (w/o local norm), we just use the global normalization across the responses in a group. In Table 3, we observe that HiPO achieves significant improvements in performance and efficiency when compared to these two variants.

**Effect of different $\gamma$ values.** Figure 4 shows that, when the value of $\gamma$ is set to 0.00, the reward for the judge token lacks information about the current response, resulting in lower model accuracy and higher token length. On the other hand, when $\gamma$ is set too high, the scales of the two terms ($\text{mean}(\mathbf{r}_{\text{off}})$ - $\text{mean}(\mathbf{r})$) and ($r_i - \text{mean}(\mathbf{r})$) become imbalanced, which leads to a decrease in model accuracy and an increase in token length.

**Effect of different rollout numbers.** Table 5 shows that, when the rollout number $N$ is set to 16, the model achieves better average performance, shorter token length, and lower think rate. We attribute this to the fact that this configuration provides sufficient data to explore diverse possibilities while avoiding excessive samples with redundant reasoning that dilute the training signal. As a result, the model focuses more on learning from higher-quality samples, leading to a more concise strategy with improved accuracy, reduced token length, and lower think rate.

**Effect of different $\omega$ values.** Table 5 shows that, setting $\omega$ to 0.01 provides a balanced trade-off between performance and efficiency. This configuration mitigates the overly conservative behavior

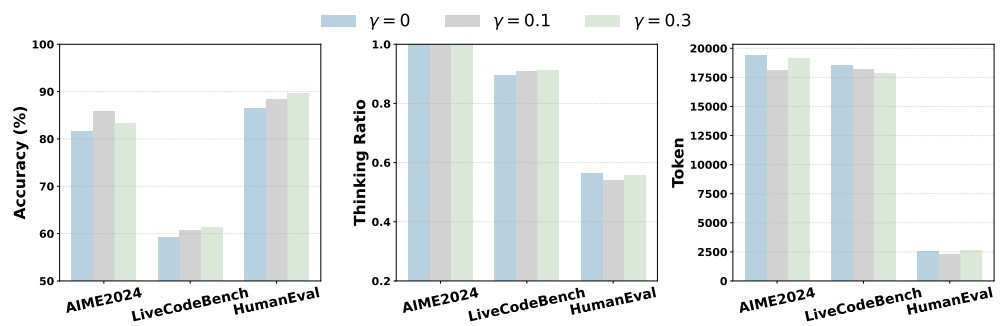

Figure 4: Performance of different $\gamma$ values.

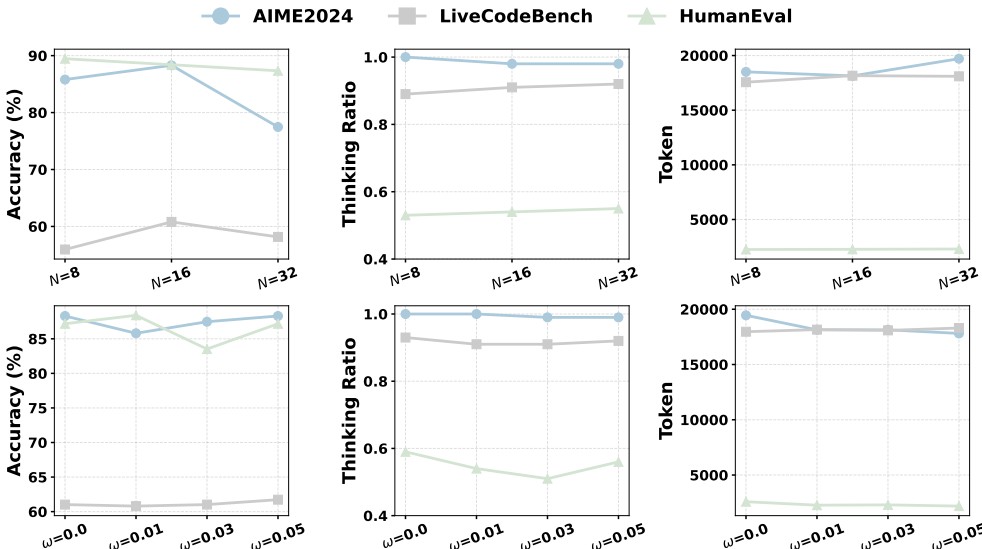

Figure 5: Performance of different rollout numbers and $\omega$ values.

seen at 0.0 while avoiding the overly aggressive behavior at higher settings, ultimately achieving the largest efficiency gains with minimal performance loss.

## 4.4 FURTHER ANALYSIS

| Method | AIME24 | | | LiveCodeBench | | | HumanEval | | | MBPP | | |
|---|---|---|---|---|---|---|---|---|---|---|---|---|
| | Acc | Length | Ratio$_T$ | Acc | Length | Ratio$_T$ | Acc | Length | Ratio$_T$ | Acc | Length | Ratio$_T$ |
| **Qwen3-1.7B** | | | | | | | | | | | | |
| Cold-Start (On) | 63.3 | 24214 | 1.00 | 33.7 | 25616 | 1.00 | 77.4 | 4172 | 1.00 | 54.6 | 8587 | 1.00 |
| Cold-Start | 65.0↑2.7% | 21039↓13.1% | 1.00−0.0% | 37.4↑11.0% | 21364↓16.6% | 0.98↓2.0% | 81.7↑5.2% | 3084↓26.1% | 0.39↓61.0% | 54.4↓0.4% | 6398↓25.5% | 0.64↓36.0% |
| HiPO | 68.3↑7.9% | 17614↓27.3% | 0.98↓6% | 44.3↑31.4% | 19358↓24.4% | 0.92↓8.0% | 86.0↑11.1% | 1973↓52.7% | 0.28↓62.0% | 62.8↑15.0% | 4330↓49.6% | 0.47↓53.0% |
| **Qwen3-32B** | | | | | | | | | | | | |
| Cold-Start (On) | 81.7 | 19551 | 1.00 | 65.4 | 17885 | 1.00 | 87.8 | 4298 | 1.00 | 76.2 | 4753 | 1.00 |
| Cold-Start | 85.0↑4.3% | 16542↓15.4% | 1.00−0.0% | 65.9↑0.8% | 14935↓16.5% | 0.87↓13.0% | 92.1↑4.9% | 2785↓35.2% | 0.47↓53.0% | 78.4↑2.2% | 3991↓16.0% | 0.51↓49.0% |
| HiPO | 88.3↑8.1% | 14873↓23.9% | 0.98↓2.0% | 68.5↑4.5% | 12721↓28.9% | 0.82↓18.0% | 92.7↑5.6% | 824↓80.8% | 0.18↓82.0% | 84.4↑10.8% | 2070↓56.4% | 0.24↓76.0% |

Table 4: Performance of HiPO on more models.

We analyze two key dimensions: (i) reasoning-mode activation (`<think_on>` vs. `<think_off>`) and (ii) token efficiency across RL training steps and benchmark tasks. Specifically, during the training and evaluation processes, we track how the model's decision-making evolves by monitoring the frequency of reasoning-mode activations and the corresponding output length.

**Think-on vs. Think-off Dynamics During Training and Inference**  We logged the frequency of `<think_on>` and `<think_off>` activations at each step. As shown in Figure 6(a), HiPO not only improves final accuracy but also sharpens the model's gating behavior, allowing it to skip unnecessary reasoning. Specifically, the gap between `<think_on>` and `<think_off>` activations decreases from 89.5% at the beginning of training to 53.1% by the end. In Figure 6(b) shows the proportion of

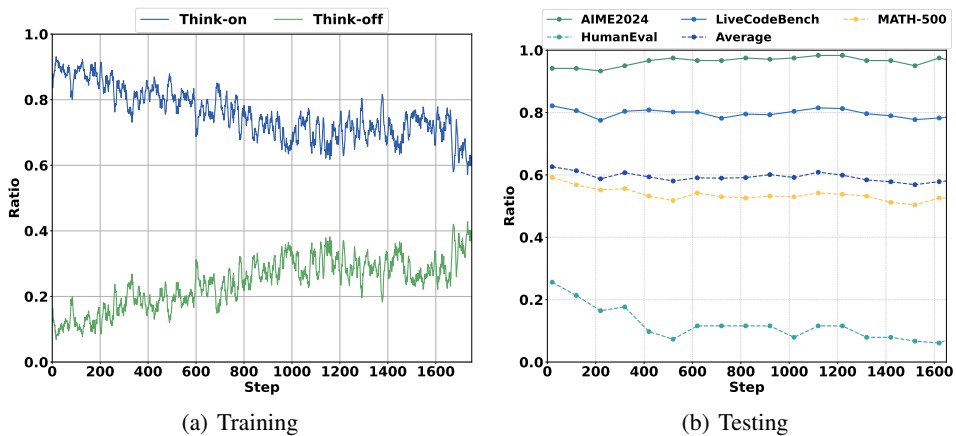

(a) Training            (b) Testing

Figure 6: (a) Think-on and Think-off ratio in training. (b) Think-on ratio of different datasets.

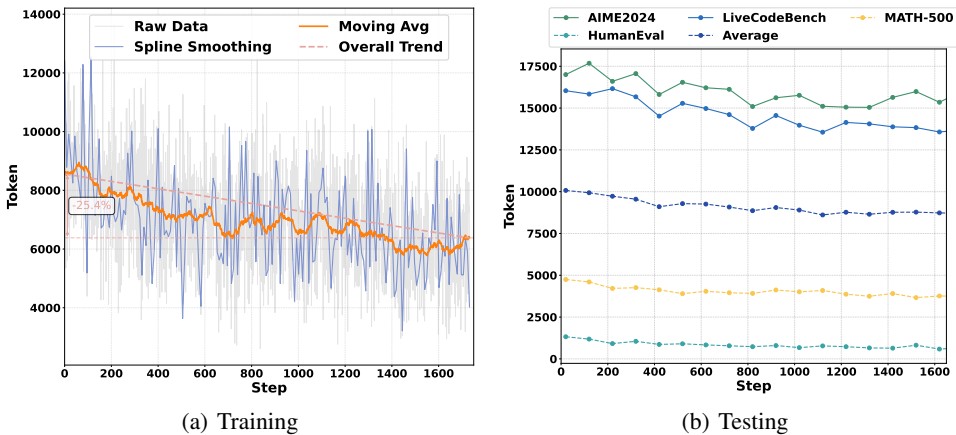

(a) Training            (b) Testing

Figure 7: (a) Average token usage in RL training. (b) Token usage of different datasets.

Think-on activations across different datasets during inference. Reasoning-intensive tasks, including AIME2024, and LiveCodeBench, consistently demonstrate high Think-on activation rates (>70%) throughout training. Conversely, tasks that require less explicit reasoning, such as HumanEval — exhibit a clear downward trend in Think-on activation as training progresses.

**Token Count Dynamics During Training and Inference** During RL training, the average token count shows a consistent downward trend in Figure 7(a), which indicates that the model gradually learns to produce more concise responses and highlight the HiPO reward design in encouraging efficient token usage Besides, Figure 7(b) shows the corresponding dynamics in average token counts per generated response during inference, and we also observe consistent token reduction in training.

**Generalization on More Models** In Table 4, we report the performance of HiPO on Qwen3-1.7B and Qwen3-32B, which shows consistent improvements on both accuracy and efficiency.

## 5 CONCLUSION

In this work, we introduced HiPO, a hybrid framework for adaptive reasoning in LLMs. By combining a hybrid data pipeline with a hybrid reinforcement learning reward system, HiPO enables models to dynamically balance Think-on and Think-off reasoning, mitigating the issue of overthinking while preserving accuracy. Experiments demonstrate that HiPO achieves competitive or superior accuracy with significantly improved token efficiency and reduced reasoning redundancy.

ETHICS STATEMENT

This research uses only publicly available datasets under their original licenses and involves no personal or sensitive information. All code and data will be released upon acceptance to ensure transparency and reproducibility. We acknowledge potential risks such as bias or misuse and encourage responsible application of our methods.

REPRODUCIBILITY STATEMENT

We have made extensive efforts to ensure the reproducibility of our work. The baseline models we evaluated are detailed in Section 4.1. We provide the implementation details of the proposed method, including the data construction pipeline in Section 3.1, the data source in Section A.3, the training procedure in Section 3.2, and the evaluation settings in Section 4.1. We will release the data source and the code upon paper acceptance to facilitate reproduction and future research by the community.

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

# A  APPENDIX

## A.1  USE OF LLMS

LLMs were used solely to assist in editing, formatting, and improving the clarity of the manuscript. All ideas, experiments, and analyses were conceived and executed by the authors. No LLM outputs were used as experimental data or results in this work.

## A.2  THE DECLINE IN QWEN3'S PERFORMANCE ON THE TEST SET.

This section demonstrates the decline in Qwen3's performance on AIME2024, AIME2025, HumanEval, and LiverCodeBench. We trained Qwen3 using AM-DeepSeek-R1-0528-Distilled, AM-Thinking-v1-Distilled, and OpenThoughts3-1.2M. The Figure 8, when the number of training steps reaches 150, Qwen3's accuracy on all benchmarks declines. Note that the batch size is set as 512 and other parameters are same as the implementation details in the main paper.

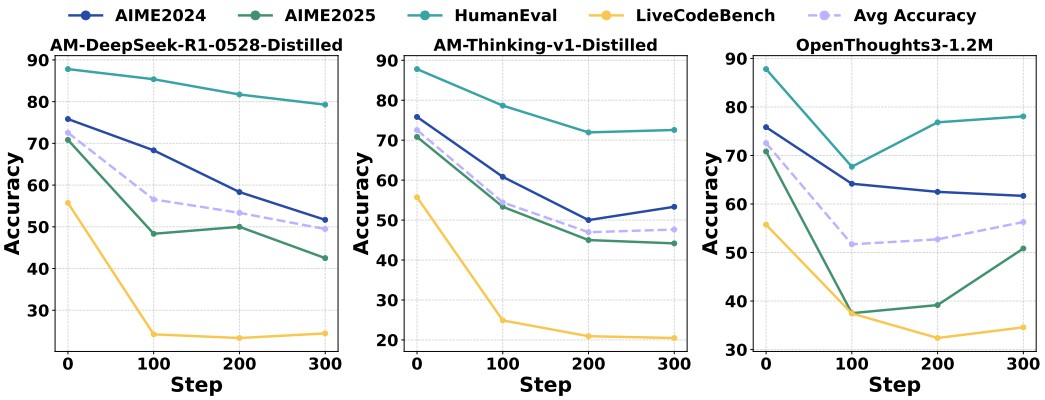

Figure 8: The decline in Qwen3's performance on the AIME2024, AIME2025, HumanEval, Live-CodeBench.

## A.3  DATA SOURCE

Our dataset is derived from several open-source reasoning corpora covering both code and mathematics. As shown in Table 5, queries come from AM-Thinking-v1-Distilled [1], II-Thought-RL [2], AceReason-Math [3], and Skywork-OR1-RL-Data [4]. This composition ensures diversity across domains and provides a reliable basis for model training and evaluation.

## A.4  PROMPT TEMPLATES

In this section, we provide the prompt templates for the response generation and judge analysis generation.

---

**Response Generation**

Please read the following question carefully and provide a clear answer.

—

Query

—

---

[1]https://huggingface.co/datasets/a-m-team/AM-Thinking-v1-Distilled

[2]https://huggingface.co/datasets/Intelligent-Internet/II-Thought-RL-v0

[3]https://huggingface.co/datasets/nvidia/AceReason-Math

[4]https://huggingface.co/datasets/Skywork/Skywork-OR1-RL-Data

| Category | Data Source | # Query |
|---|---|---|
| Code | AM-Thinking-v1-Distilled (Tian et al., 2025) | 85k |
| | II-Thought-RL (Internet, 2025) | 20k |
| Math | AceReason-Math (Chen et al., 2025b) | 49k |
| | AM-Thinking-v1-Distilled (Tian et al., 2025) | 32k |
| | II-Thought-RL (Internet, 2025) | 30k |
| | Skywork-OR1-RL-Data (He et al., 2025) | 24k |

Table 5: Description of data sources.

---

**Judge Analysis Generation**

You are tasked with analyzing the characteristics of a question to determine why it **requires** complex reasoning.

Your should **not** attempting to answer or infer its solution.

You should analyse user's question to determine the **core task intention**—that is, what the user wants the model to do. (e.g., write and validate code based on a problem description, etc.).

Then briefly outline the basic approach to accomplishing this task (e.g., write SQL code to retrieve imformation, etc.).

Based on the required approach, assess the **reasoning complexity**, and indicate whether it involves multiple steps or deep analysis. Do not solve the question or provide an answer. Focus solely on interpreting the task type, approach, and cognitive demand.

Be concise: your analysis must be no more than two lines and under 500 characters. Use clear, natural, and varied language. End your explanation with a statement indicating that complex reasoning is required (Think-on), but express this conclusion with a natural and diverse phrase, not repeating any single pattern. The meaning must be clear, but the expression can vary.

Please analyze the following question as required above:

—

Model Response

—

