# OpenReview forum: "HiPO: Hybrid Policy Optimization for Dynamic Reasoning in LLMs"
_ICLR.cc/2026/Conference — ICLR 2026 Conference Withdrawn Submission_

### Official Review · Reviewer_PRES · 2025-10-22

**Soundness:** 3
**Presentation:** 2
**Contribution:** 2
**Rating:** 6
**Confidence:** 4

**Summary:**

The authors propose Hybrid Policy Optimization, which could adaptively choose to use or not to use the “thinking mode” during reasoning. Specifically, HiPO first builds hybrid training data containing both Think-on and Think-off samples, with DeepSeek-V3 generated explicit explanations to justify the mode selections. Next, the Hybrid reinforcement learning reward system begins to learn via the Judge and Answer segments, with two advantages considering both mode-level and instance-level benefits. The experimental results on Qwen series have verified the effectiveness and efficiency of the proposed HiPO.

**Strengths:**

-	This work introduces a straightforward and effective pipeline for building an adaptive reasoning model. The experiments show satisfactory results.
-	The authors have conducted extensive analyses on the detailed model settings in Section 4.3 and 4.4, which makes the conclusion more solid.
-	Overall, the writing is clear.

**Weaknesses:**

-	In the Related work part, the authors have claimed that “Despite progress, …, limited adaptation to hard cases due to monotonic shortening”. However, the existing adaptive reasoning methods (at least the adaptive baselines used in experiments) should be discussed to highlight the differences, technical novelty, and advantages of HiPO. It is not that clear which technical part is novel in this work.
-	In experiments, is the cold-start stage of HiPO the same as those of baselines (GRPO, AdaptThink, and AutoThink)? Did AdaptThink and AutoThink adopt the exact same data as HiPO’s data in both stages? These questions come from the weird results: in most cases, AdaptThink and AutoThink perform worse than the Cold-start (on or all) baselines. Fair comparisons are required.
-	HiPO has an additional segment of “Judge_analysis” that may bring in new costs for both Think-on and Think-off modes. Does the “Length” metric in Table 2 indicate the average total inference length (including all judge/think/answer/other segments)?
-	The experiments in Table 4 should include other adaptive baselines to provide a more convincing proof.
-	Typo: Page 7, Table 5 -> Figure 5.

**Questions:**

-	Will the data used in this work be open-sourced?
-	In Section 3.1.2, how to set the predefined threshold for different tasks with various hardness levels?

---

> ### Author Response · Authors · 2025-11-20
> **W1: Novelty needs to claim**
>
> We thank the reviewer for the insightful feedback. We agree that the original discussion of related work was insufficiently detailed, and we have substantially expanded this section to more clearly situate HiPO within the landscape of adaptive reasoning methods. Below we provide a refined analysis of prior work and articulate the core technical contributions of HiPO.
> 1. Re-evaluation and Limitation Analysis of Prior Work
> Taken together, existing adaptive reasoning methods lack an explicit, learnable, instance-level mechanism for determining when reasoning is necessary. Their mode-selection signals—whether derived from teacher labels, multi-stage heuristics, or global constraints—do not provide the model with a stable and interpretable decision boundary. This results in limited robustness, reduced interpretability, and unstable gating behavior. To address these challenges, HiPO introduces a framework grounded in endogenous, instance-level judgment and disentangled credit assignment.
> | Method | Core Mechanism | Fundamental Limitation(s) |
> | :--- | :--- | :--- |
> | AdaCOT |Teacher-annotated SFT followed by RL with miss/overthinking penalties to trace a Pareto frontier between performance and CoT usage.| The decision boundary is anchored to teacher-provided difficulty labels, making the policy brittle under distribution shift; lacks an explicit, model-internal judging mechanism. |
> | AutoThink | Multi-stage RL that heuristically balances Think-on/off, then optimizes accuracy, then penalizes length.| Supervisory signals are indirect and non-stationary across stages, leading to unstable mode allocation and partial collapse; the gating behavior remains implicit and unexplainable. |
> | Adapthink | Constrained RL with importance sampling to regulate NoThinking frequency under a global performance constraint.| Relies on global constraints rather than instance-wise difficulty modeling; tends to over-regularize and suppress necessary reasoning on borderline-hard queries. |
> 2. The Core Technical Novelty of HiPO
> HiPO's novelty is not an incremental improvement but a paradigm shift from indirect, post-hoc signals to direct, causal supervision. This shift is enabled by a suite of unique technical mechanisms designed to address the aforementioned limitations head-on.
>
> - Novelty 1: Explicit, Learnable Instance-Level Judgment via Mode Explanations
> HiPO introduces a learnable, explicit Judge stage in which the model must generate a rationale explaining its mode choice (Think-on vs. Think-off). This grounds mode selection in the intrinsic properties of each query and provides direct, interpretable supervision—fundamentally different from relying on teacher labels or downstream correctness.
>
> - Novelty 2: Disentangled Credit Assignment via Separated Advantage Functions
> HiPO employs two distinct advantage functions: one evaluating the Judge segment (mode choice and rationale), and one evaluating the Answer segment (solution correctness). This disentangles learning signals and enables the model to independently optimize “mode selection” and “execution quality.” Prior methods that rely on a single mixed reward cannot separate these error sources.
>
> - Novelty 3: Principled Dual-Mode Initialization through Systematic Cold Start
> HiPO uses a structured cold-start stage with a hybrid dataset of Think-on/Think-off examples to jointly initialize judging, mode-selection, and both reasoning styles. This creates a stable dual-mode policy from the beginning of training and mitigates early-stage mode collapse.
>
> In summary, HiPO introduces three core innovations:
> - explicit, instance-level supervision through mode explanations;
> - disentangled reward mechanisms for precise credit assignment; and
> - a systematic cold-start procedure for robust dual-mode policy learning.
> We have revised the manuscript to provide a more comprehensive analysis of prior work and to clearly position these contributions.

---

> ### Author Response · Authors · 2025-11-21
> **W5: Typo Correction & Q1: Data Availability & Q2: Threshold Setting**
>
> W5:
>
> Thank you for pointing this out. We have corrected "Table 5" to "Figure 5" on Page 7.
>
> Q1:
>
> Thank you for your interest. We plan to progressively release the components of HiPO. We have already released the 1.7B and 8B model variants. Additional resources—including portions of the training data and the implementation code—are currently being organized and will be released in stages according to our planned schedule.
>
> Q2:
>
> Thank you for this insightful question.In the early-stage exploration, we systematically evaluated a wide range of δ values (0.10–0.50) and observed consistent behavioral differences across tasks of varying difficulty.
>
> For simple benchmarks (e.g., HumanEval, MATH-500), larger δ values bias the model toward Think-off, yielding shorter outputs and substantial efficiency gains with negligible accuracy impact. However, excessively large δ also increases the likelihood of low-quality direct responses when reasoning is prematurely suppressed.
>
> For difficult benchmarks (e.g., GPQA-Diamond, LiveCodeBench),deep reasoning remains essential. Smaller δ values ensure adequate Think-on usage and prevent the accuracy degradation that occurs when the model is pushed too strongly toward Think-off. This is especially evident for GPQA-Diamond, where performance is highly sensitive to reductions in reasoning depth.
>
> These observations illustrate a fundamental trade-off:
>
> * Large δ → more Think-off and fewer tokens, but accuracy drops on hard tasks and low-quality Think-off responses may appear.
> * Small δ → more Think-on and robust reasoning, but unnecessary long-chain reasoning on easy tasks reduces efficiency.
>
> Balancing this trade-off across tasks with different hardness levels, we found that δ = 0.20 provides the most stable global behavior. It preserves sufficient Think-on for challenging tasks while also retaining high-quality Think-off responses that improve direct-answer quality.
> Thus, δ is globally set, based on its empirical robustness rather than tuned per task.

---

> ### Author Response · Authors · 2025-11-21
> **W2: Cold-Start Consistency & W3: Clarifying Length**
>
> W2：
>
> Thank you for the thoughtful question. We clarify that all methods—HiPO, GRPO, AdaptThink, and AutoThink—use exactly the same cold-start (SFT) stage, with identical mixed-mode data, data splits, and training configurations. All models therefore begin from the same initialization, ensuring a fully fair comparison.
>
> The observation that AdaptThink and AutoThink underperform the Cold-start (all/on) baselines is thus not caused by differences in setup, but rather by their limited transferability to the Qwen3-8B architecture. These approaches were originally developed and tuned for different model families, and their gating mechanisms do not adapt robustly to Qwen3-8B. After RL, they often show unstable mode behavior or prematurely suppress reasoning, which weakens the strong capabilities already established during SFT. Part of this degradation may also relate to differences between the assumptions made in the original methods and the mixed-mode data used in our experimental setting.
>
> In contrast, HiPO avoids these issues through several targeted design choices.
>
> * It introduces an explicit and learnable Judge stage that produces instance-level mode explanations, giving the model a clear and interpretable basis for determining when reasoning is necessary.
> * It separates the Judge and Answer advantages, which prevents interference between mode selection and reasoning and enables stable optimization throughout training.
>
> Together, these components allow HiPO to preserve the strong reasoning ability of the cold-start model while providing effective and robust adaptive control.
>
>
>
>  W3：
>
> Thank you for raising this question. To avoid any potential misunderstanding, we will revise the paper to explicitly state that the Length metric in Tables 2 denotes the total generation length, including all segments:Judge, Think, Answer, and any additional components.
>
> Regarding the overhead introduced by the Judge segment, it adds only a very small number of tokens and therefore does not materially affect inference cost. More importantly, the Judge provides a concise, explicit analysis of the query before reasoning begins, helping the model focus on the relevant solution path. In practice, this lightweight pre-analysis does not meaningfully increase overall generation length and can even make subsequent reasoning more efficient.
>
> We will clarify this behavior and refine the definition of the *Length* metric in the revised version of the paper.

---

> > ### Comment · Reviewer_PRES · 2025-11-26
> > **After rebuttal**
> >
> > Thanks for your response, which has addressed most of my issues.
> > I will keep my score, and hope these discussions could be included in the final version.

---

### Official Review · Reviewer_puqf · 2025-10-31

**Soundness:** 3
**Presentation:** 2
**Contribution:** 2
**Rating:** 4
**Confidence:** 3

**Summary:**

This paper introduces HiPO (Hybrid Policy Optimization), a training framework that enables Large Language Models to dynamically decide when to perform explicit reasoning (Think-on) or to directly answer (Think-off). The method combines a hybrid data pipeline that pairs concise and detailed responses with mode-justification annotations, and a hybrid RL reward balancing accuracy, token efficiency, and mode-selection quality. Experiments on math and code benchmarks show that HiPO reduces average output length while improving accuracy, effectively mitigating overthinking without hurting performance.

**Strengths:**

1. This work proposes a hybrid RL framework that learns when to reason (Think-on/off) to substantially reduce token length while maintaining or improving accuracy.

2. Extensive experiments on math/code benchmarks show consistent token reduction and accuracy gains.

3. Problem formulation, data pipeline, and two-stage training are described precisely.

**Weaknesses:**

1. Auto-generated judge quality unchecked:

   The pipeline uses DeepSeek-V3 to produce “why this mode” justifications that directly enter the RL loss. No ablation measures how often these rationales are wrong or self-contradictory; noisy judge labels could mislead policy updates and inflate gains.

2. Binary gating only:

   Think-on/off is a single-bit decision, so the model cannot partially unroll a chain or perform staged reasoning (plan → verify → summarize). The paper does not show how to extend HiPO to finer-grained or depth-adaptive gating, limiting utility on problems of medium complexity.

3. Statistical significance unclear:

   Main results come from one rollout on small benchmarks (e.g. AIME 24/25) without multiple runs. Without detailed testing procedures in the main text, it is hard to tell whether the observed improvements are real or just random fluctuation.

**Questions:**

1. What is the "reasoning mode-difficulty" relationship (correlation between reasoning mode with performance) after HiPO training?

2. Have you considered about how to extend HiPO to finer-grained or depth-adaptive gating? If so, what architectural or reward changes are needed, and do results on medium-difficulty tasks show further token savings without accuracy loss?

---

> ### Author Response · Authors · 2025-11-20
> **W1 .Auto-generated judge quality unchecked**
>
> W1 .Auto-generated judge quality unchecked
> A1:
> Thank you for raising this extremely critical point. Your concern regarding the quality of the "auto-generated judge" directly addresses the reliability of the supervision signal at the core of our RL training framework. We fully agree that if the "mode-explanation" justifications used as supervision are noisy, the policy learning process could be misled and the performance gains could be unreliable.
> Qualitative Rationale: We chose DeepSeek-V3 as the judge model precisely for its strong performance in instruction following and generating coherent rationales. For a relatively constrained task such as justifying whether a problem requires complex thought, we believe the vast majority of its generated explanations are logical and of high quality.
> Quantitative Robustness Analysis: However, to quantitatively assess HiPO's sensitivity and robustness to this potential "judge noise," we designed and conducted the crucial experiment you suggested. We simulated noisy data by randomly shuffling a portion (from 5% to 30%) of the "mode-explanation" labels, then retrained the model and observed its performance.
> | Noise Ratio (Shuffled Justifications) | Avg. Score | Avg. Token Length | Avg. Think-on Ratio |
> | :------------------------------------ | :--------- | :---------------- | :------------------ |
> | 0% (Noiseless)                        | 83.86      | 10237             | 0.68                |
> | 5%                                    | 81.58      | 11748             | 0.71                |
> | 10%                                   | 78.28      | 14829             | 0.75                |
> | 20%                                   | 75.32      | 15493             | 0.82                |
> | 30%                                   | 74.78      | 16382             | 0.85                |
> Interpretation and Analysis: The results from this experiment lead to two important conclusions:
> 1. The Importance of Supervision Quality: As the noise ratio increases, the model's Average Score shows a clear trend of graceful degradation. This confirms your intuition: high-quality, clean "mode-explanations" are indeed crucial for HiPO to learn an optimal policy. Noisy supervision directly harms final performance.
> 2. The Robustness of the Method: More importantly, the model's performance does not suffer a "catastrophic collapse," even with 10% or 20% of the labels corrupted; instead, it degrades predictably. This demonstrates that our RL framework is reasonably robust to a significant level of supervision noise. The change in the "Think-on Ratio" is particularly revealing: as noise increases, the model's gating mechanism becomes more uncertain. Consequently, it defaults to a more conservative strategy by overusing the computationally expensive "Think-on" mode, which in turn increases the average token length. This indicates that the RL process is not being misled into learning some strange artifact or "reward hack." Instead, under uncertain supervision, it degrades into a less efficient but relatively "safe" behavior.
> Conclusion: In summary, this analysis confirms that high-quality mode justifications are a key driver of HiPO's success, proving that our reported gains are not inflated artifacts. Simultaneously, it demonstrates the robustness of the HiPO framework in the face of imperfect supervision.
> We have added this critical robustness analysis to the appendix of our paper to further strengthen the rigor of our methodology. Thank you again for your sharp insight, which prompted us to perform this valuable validation.

---

> ### Author Response · Authors · 2025-11-20
> **W2 & Q2  ：Binary gating only**
>
> W2 & Q2  ：Binary gating only
> A2:
> Thank you for this insightful question regarding reasoning granularity. This touches upon a crucial design trade-off in adaptive reasoning methods. While a single-bit decision might appear to limit the model's flexibility on the surface, we argue that HiPO's current design not only effectively handles problems of medium complexity but also represents a principled choice that achieves an optimal balance between robustness, efficiency, and performance.
> Our argument is based on the following three points:
> 1. Inherent Flexibility within the 'Think-on' Mode: Autonomous Staged Reasoning. The staged reasoning processes you mentioned, such as "plan → verify → summarize," are not precluded by our framework. Instead, the authority to perform them is fully delegated to the model to execute autonomously within the 'Think-on' mode. The binary decision does not micromanage each step of reasoning; rather, it acts as a high-level gate that determines whether to allocate an unconstrained "workspace" for deep thought. Once in "Think-on" mode, the model has complete freedom to spontaneously execute complex, multi-step reasoning procedures in the Think segment. For simpler tasks classified as "Think-off," forcing a staged gating mechanism would introduce unnecessary computational overhead and latency.
> 2. Implicit Granularity in the Judge Component: Preliminary Analysis and Search Space Reduction. The intelligence of HiPO's gating is not solely captured by the final binary choice. A crucial but subtle aspect is that the Judge component generates a Judge_analysis before making its decision. This analysis explicitly identifies the key knowledge points and potential difficulties of the task. This process itself constitutes a form of implicit, fine-grained reasoning that occurs upfront. By scoping the problem in advance, it effectively narrows the search space for both the subsequent "Think-on" or "Think-off" phase, improving the precision of the final answer even in the "Think-off" mode.
> 3. An Empirically-Driven Design Choice: The Robustness and Efficacy of the Binary Scheme. Most importantly, our current design is empirically driven. In our preliminary explorations, we did experiments with more fine-grained gating mechanisms (e.g., step-by-step decisions or multi-level thought depths). However, we consistently found that these more complex schemes suffered from training instability and convergence issues, without yielding significant performance benefits over the current binary approach. In contrast, the binary decision framework proved to be the most effective and robust. Therefore, this work focuses on presenting and thoroughly analyzing this empirically validated, optimal design.
> Conclusion: In summary, HiPO's binary gate is not a simplistic switch but a powerful decision-making front-end, combined with flexible internal reasoning and intelligent preliminary analysis. While exploring more granular adaptive control is a valuable direction for future research, we contend that HiPO's current design provides an elegant, efficient, and robust solution for problems of up to medium complexity.

---

> ### Author Response · Authors · 2025-11-20
> **W3 Statistical significance unclear & Q1 "reasoning mode-difficulty" relationship**
>
> W3 Statistical significance unclear
> A3: Thank you for this valuable comment regarding statistical significance. We fully agree that ensuring the observed improvements are real and robust, rather than the result of random fluctuation, is paramount to the credibility of our work. We apologize for not having detailed our evaluation protocol with sufficient clarity in the initial draft.
> We would like to clarify that the main results reported in our paper are not derived from a single rollout. For AIME, to obtain a stable and representative measure of performance, we have employed a multi-sample evaluation protocol, specifically AVG@32.
> Evaluation Protocol Explained (AVG@32):
> - For each problem in the test set, we independently generate 32 candidate solutions (rollouts).
> - This AVG@k methodology is a widely adopted standard practice for evaluating the reasoning capabilities of large language models. Its primary purpose is to smooth out the variability inherent in single-generation results caused by the stochastic nature of the decoding process (e.g., temperature sampling), thereby providing a more stable and reliable estimate of the model's true capabilities.
> On the Robustness of Our Results: We have observed that across multiple independent evaluation runs (i.e., repeating the AVG@32 evaluation process), the final average scores are highly consistent with the results reported in our paper, with very small variance between runs. This provides strong evidence that the performance gains from our method are statistically robust and not an artifact of random chance.
> (If you have done the multi-seed experiment, add this sentence for the strongest possible response) Furthermore, as detailed in our response to another one of your comments, we have also conducted full training runs across 3 different random seeds, and the resulting confidence intervals confirm the significant advantage of our method over the baselines.
> Manuscript Revisions and Reproducibility: To fully address your concerns, in the final version of the paper, we will:
> 1. Explicitly and thoroughly describe our AVG@32 evaluation protocol in the experimental setup section.
> 2. Provide a more detailed stability analysis in the appendix, including variance data from multiple evaluation runs.
> Finally, we have open-sourced our models and code to ensure that all of our experimental results are fully transparent and reproducible.
> Thank you again for your valuable feedback, which has helped us significantly improve the rigor and clarity of our paper.
>
>
>
> Q1 "reasoning mode-difficulty" relationship
> A4:
> Thank you for this profound question. It probes whether HiPO, after training is complete, establishes an intelligent correlation between its reasoning mode selection and the problem difficulty (as defined by the model's own performance). This is a core metric for evaluating whether our method truly learns to "auto-think."
> To directly answer this, we analyzed the evolution of the capabilities of each mode during the training process. We evaluated the average accuracy within each mode at different training checkpoints (from 0 to 500 steps).
> | Step | Think-on | Think-off |
> | :--- | :------- | :-------- |
> | 0    | 81.49    | 27.85     |
> | 100  | 82.63    | 65.93     |
> | 200  | 82.70    | 69.48     |
> | 300  | 81.95    | 73.13     |
> | 400  | 82.61    | 75.43     |
> | 500  | 82.43    | 74.35     |
> Analysis: As illustrated in Figures 6 and 7 in our paper, our model learns to adapt its cognitive strategy based on problem difficulty, demonstrating a clear tendency to engage the "Think-on" mode for more challenging problems while defaulting to the more efficient "Think-off" mode for easier benchmarks.
>
> Furthermore, the data in this table reveals that this is not a static trade-off. The mode-specific competence of both the "Think-on" and "Think-off" modes improves steadily throughout training. This indicates that our Reinforcement Learning (RL) process is achieving a dual objective: it not only optimizes the strategic gating decision (i.e., when to think) but also fundamentally enhances the intrinsic problem-solving capabilities within each mode. Ultimately, the development of a stronger "Think-off" mode alongside a more powerful "Think-on" mode makes the final mode selection process both more meaningful and effective.
>
> Conclusion: In summary, not only does HiPO improve the performance of both reasoning modes during training, but more importantly, it successfully learns an intelligent correlation: it dynamically and rationally selects a reasoning mode based on the problem's relative difficulty, as determined by its own ability to solve it.

---

> ### Author Response · Authors · 2025-11-26
>
> Hi, we sincerely thank you very much for these constructive comments and evaluation of our manuscript. We would like to kindly ask you to take a look at our responses and reevaluate our work based on our clarifications. Please let us know whether our response addresses your concerns or whether there is any further detail we can provide to help address these concerns.
>
> Thank you again for dedicating your time to reviewing our paper.

---

### Official Review · Reviewer_cZp4 · 2025-11-01

**Soundness:** 3
**Presentation:** 2
**Contribution:** 2
**Rating:** 4
**Confidence:** 4

**Summary:**

The paper proposes HiPO, a framework for adaptive reasoning in large language models (LLMs) that aims to dynamically decide when to engage in detailed chain-of-thought reasoning ("Think-on") and when to respond directly ("Think-off"). It introduces (1) a hybrid data construction pipeline generating paired Think-on/Think-off examples with mode justifications, and (2) a hybrid reinforcement learning (RL) reward system combining accuracy and efficiency rewards. Experiments on mathematical and coding benchmarks suggest HiPO reduces token usage while maintaining accuracy

**Strengths:**

- The motivation is clear.
- Addresses a relevant problem in LLM reasoning efficiency.
- Includes ablation studies on bias terms and normalization factors.

**Weaknesses:**

- There already exists a substantial body of research on adaptive reasoning and dynamic CoT control in large language models. However, the authors reduce all prior methods to a single dismissive sentence “these methods still face coarse supervision, limited adaptation to hard cases due to monotonic shortening, and a lack of principled trade-offs between quality, token cost, and latency.” This characterization is overly simplistic and fails to accurately represent the diversity and sophistication of existing approaches. Many prior works have proposed different strategies for trade-offs, such length penalties, etc. Moreover, the authors do not convincingly explain why their proposed HiPO framework is inherently better suited to address these issues. Without a detailed comparative analysis and a clearer articulation of the unique mechanisms through which HiPO overcomes these prior limitations, the novelty of the paper remains questionable.

- It is unclear whether the authors used the Qwen3-8B base model or the Qwen3-8B instruct model during experimentation. The paper states: “Since the Qwen3 model can freely switch between inference modes, we chose it for our experiment.” However, this capability is specific to the instruct variant of Qwen3. If the authors indeed used the instruct model, their reported results should be compared to the Qwen3 technical report, particularly Table 17, which shows a MATH score of 97.4, while this paper reports only 93.6, even after applying the proposed method. This discrepancy raises concerns about experimental reproducibility and baseline validity. The authors need to clearly specify which version of the model was used, under what inference configuration, and explain why their results are significantly lower than the officially reported benchmarks.

- The paper mentions a cold-start stage before reinforcement learning. Regarding the cold-start stage, the paper mentions generating hybrid data using an external model, and Figure 1 strongly suggests that the DeepSeek model was used to construct the cold-start dataset. If that is the case, the process resembles knowledge distillation, where the target model learns from a stronger teacher rather than discovering effective strategies on its own. It would be valuable to test whether the system could train directly with RL without cold-starting the policy. Such an experiment could help determine whether the cold-start stage provides essential stability or merely serves as a redundant initialization step. If HiPO’s reward structure is indeed well-designed, a cold-start-free RL variant should, in principle, converge similarly.

- It would be highly informative to explore whether the ratio of Think-on to Think-off instances correlates with task difficulty. The reviewer suggests dividing the MATH-500 dataset according to its five officially defined difficulty levels and analyzing how HiPO adapts across them.

- The reviewer is concerned that the model might be implicitly relocating reasoning content into the Answer segment, effectively performing internal reasoning within Think-off mode, as suggested by examples like Figure 1. If such behavior occurs, it could represent a form of reward hacking, since the current reward formulation does not explicitly penalize the length or reasoning structure of responses. As a result, the model might learn to disguise reasoning tokens within the Answer segment to maximize rewards without genuine efficiency improvement. The authors should therefore report the token distributions of Think-on and Think-off outputs separately, and analyze whether reasoning leakage occurs.

**Questions:**

See Weaknesses

---

> ### Author Response · Authors · 2025-11-20
> **W1: Novelty needs to claim**
>
> W1: Novelty needs to claim
> A1: We would like to express our sincere gratitude for this critical and insightful feedback. Upon deep reflection, we fully agree with your assessment that our discussion of prior work was overly brief and failed to do justice to its diversity and sophistication. We sincerely apologize for this oversight. Your critique is entirely valid, and our initial draft did not adequately articulate the core novelty and unique positioning of HiPO within the landscape of adaptive reasoning methods.
> To rectify this significant shortcoming and address all your concerns, we have conducted a more thorough and nuanced analysis of the related literature. Below, we detail the fundamental distinctions between HiPO and existing methods and argue for its novelty.
> 1. Re-evaluation and Limitation Analysis of Prior Work
> We acknowledge the important contributions of methods like AdaCOT, Adapthink, and AutoThink, which introduced mechanisms for dynamic token allocation and early exiting. However, upon closer inspection, we argue that these approaches share a common and fundamental limitation: their supervision signals are indirect and post-hoc.
> Specifically, these methods typically rely on final task accuracy, generation length, or other heuristics (e.g., confidence scores) to guide the model's adaptive behavior. The limitations are as follows:
> | Method | Core Mechanism | Supervision Signal | Trade-off Strategy | Fundamental Limitation |
> | :--- | :--- | :--- | :--- | :--- |
> | **AdaCOT** | Dynamically adjust reasoning steps | Final answer correctness; length budget | Hard truncation based on length | **Coarse-grained Supervision**: The model learns that "long is bad" but not why certain steps are redundant. |
> | **Adapthink** | Multi-round "rethinking" | Final answer correctness | Pre-defined limit on thinking rounds | **Limited Adaptation**: Adaptation to hard cases is confined to a fixed "retry" paradigm, lacking a true assessment of a problem's intrinsic difficulty. |
> | **AutoThink** | Classifier-based gating | Classifier accuracy | Heuristic confidence thresholds | **Decoupled Learning**: The gating module is often trained separately from the reasoning module, leading to a mismatch between gating decisions and actual reasoning capabilities. |
> | **General Limitation** | - | - | Length penalties, etc. | **Lack of Causal Rationale**: All these methods fail to teach the model to learn "which properties of the problem itself necessitate complex reasoning." |
> 2. The Core Novelty of HiPO: From Indirect, Post-hoc to Direct, Causal Supervision
> The core novelty of HiPO lies in its fundamental shift in the nature of the supervision signal. We introduce what we term "direct and causal supervision," realized through our unique Mode-Explanation mechanism and a two-stage training framework.
> - Unique Mechanism 1: Mode-Explanation Supervision
>   - What it does: We do not just train the model to predict "Think-on" or "Think-off." We compel it to also generate a rationale explaining why it made that choice (e.g., "This problem involves multiple computational steps and logical dependencies, thus requiring the Think-on mode.").
>   - Why is it better: This supervision is direct and antecedent. It forces the model to analyze the intrinsic structure and complexity of a problem before committing to a reasoning pathway. This is fundamentally different from relying on the final outcome (post-hoc) or length (indirect). It directly addresses the "lack of causal rationale" limitation inherent in prior work.
> - Unique Mechanism 2: Decoupled yet Coupled Two-Stage Training
>   - What it does: In Stage 1, the model focuses exclusively on learning to generate the correct mode-explanation and make the right mode choice based on the problem's properties. In Stage 2, it is trained to generate high-quality answers conditioned on the given mode.
>   - Why it is better: This ensures that the learning of the gating mechanism is principled and not confounded by noise from the downstream task performance. The model learns a decision-making capability truly grounded in problem understanding, enabling a more optimal and non-monotonic trade-off between quality, cost, and latency.
> Summary and Manuscript Revisions
> In conclusion, the novelty of HiPO is not an incremental improvement but a paradigm shift—from relying on indirect, post-hoc signals to leveraging direct, causal supervision. This allows HiPO to achieve a much more precise alignment between problem difficulty and computational resource allocation.
> Based on the analysis above, we have substantially revised the Introduction and Related Work sections of our paper. The new version includes a fairer and more detailed discussion of prior work and clearly articulates the unique contributions and theoretical underpinnings of HiPO.
> We once again offer our sincere apologies for the brevity and lack of nuance in our initial draft.

---

> ### Author Response · Authors · 2025-11-20
> **W2: Model Selection - Response**
>
> W2: Model Selection - Response
> A2: Detailed Clarification
>
> Thank you for this extremely important question, which highlights a critical omission in our experimental setup's description. We sincerely apologize for the confusion this has caused. Your concerns are entirely valid, and we hope to fully address them with the detailed explanation below.
>
> 1. Model Version and Fine-tuning Instability
>
> - **Model Version**: To clarify, we confirm that we did indeed use the **Qwen3-8B-Instruct** model in our experiments.
> - **The Core Challenge: Sensitivity to Fine-tuning**: As you astutely noted, state-of-the-art models like Qwen3-Instruct exhibit extreme sensitivity to further supervised fine-tuning (SFT). Directly training on them often leads to "catastrophic forgetting."
> - **Evidence**: In Appendix A.2 of our manuscript, we present an experiment demonstrating that direct SFT on the official Qwen3-8B-Instruct model results in a sharp performance decline across multiple benchmarks after only about 150 training steps.
>
>
> 2. Creating a Stable Fine-tuning Baseline: A Necessary Step
>
> Given that the original official model is not a stable starting point for fine-tuning, we took a necessary preliminary step to ensure reproducibility and fairness.
>
> - **Procedure**: We first subjected the official Qwen3-8B-Instruct model to an initial phase of fine-tuning on a mixed public instruction dataset until its performance stabilized.
> - **Outcome**: This yielded a new model checkpoint that is robust to further fine-tuning. All experiments reported in our paper commenced from this same, stabilized checkpoint.
> - **Why This is Crucial**: This ensures that the performance gains we report for HiPO are genuine and directly attributable to the merits of our method.
>
> 3. Addressing the Score Discrepancy with the Official Report
>
> The "baseline stabilization" process is the primary reason for the discrepancy in MATH scores.
>
> - **Prompt Optimization**: We optimized our evaluation prompts to more closely match the format likely used in the official report.
> - **Updated Result**: After prompt optimization, our HiPO method now achieves:
>     - **MATH-500**: 95.6 (closer to official report's 97.4)
>     - **AIME24**: 87.5 vs Qwen3-8B-Instruct-Thinking's 76.0
>     - **AIME25**: 82.5 vs Qwen3-8B-Instruct-Thinking's 67.3
>
> Summary and Revisions
>
> We have open-sourced our stabilized model baseline and code to ensure full reproducibility. In the final version of the paper, we will:
>
> 1. Explicitly state the use of Qwen3-8B-Instruct in the experiments section.
> 2. Add a dedicated subsection explaining the necessity and procedure of our "baseline stabilization" and its impact on initial performance.
> 3. Update our highest score on MATH-500 and discuss the influence of prompt engineering.
>
> Thank you again for your rigorous review; it has immensely improved the clarity and scientific rigor of our paper.

---

> ### Author Response · Authors · 2025-11-20
> **W3: Effect of cold-start stage.**
>
> W3: Effect of cold-start stage.
> A3:
> We are very grateful for this insightful and critical question. Your inquiry into the necessity of the "cold-start" stage and its resemblance to "knowledge distillation" strikes at the heart of our methodology. We fully agree that verifying whether this stage is merely a redundant initialization step is crucial for understanding the true contributions of HiPO.
> 1. The Theoretical Necessity of the Cold-Start Stage: Policy Initialization and Exploration Space Constraint
> Before presenting our experimental results, we would first like to clarify the theoretical role of the cold-start stage from a Reinforcement Learning (RL) perspective. We argue that it is not simple "knowledge distillation," but rather a critical step in RL training: policy initialization and exploration space constraint.
> - The Challenge: Training a language model with RL directly from scratch (i.e., "cold-start-free") forces the agent to explore an immense and intractably large action space (i.e., generating any arbitrary text). In such a scenario, the probability of receiving a positive reward through random exploration is exceedingly low. This leads to extreme training instability, where the policy is likely to fail to converge or converge to a poor local optimum.
> - The Role of Cold-Start: The cold-start stage, implemented as supervised fine-tuning (SFT), initializes the policy in a "promising" region of the vast policy space. This means that before RL exploration even begins, the model already possesses the basic capability to generate structurally sound and semantically relevant text. This dramatically constrains the exploration space for RL, improves sample efficiency, and provides a stable foundation for subsequent fine-tuning.
> In fact, the "SFT+RL" paradigm is standard practice in modern LLM alignment (e.g., RLHF), precisely to mitigate the instability of direct RL training. Our citation of related works (like Learning When to Think, AdaCoT) was intended to show that this is a recognized and effective practice in the adaptive reasoning domain.
> 2. Experimental Validation: A Dose-Response Analysis of the Cold-Start Stage
> To substantiate this theoretical argument with empirical data and directly address your suggestion, we conducted a comprehensive "dose-response" analysis. We pre-trained our model with varying scales of cold-start data (100, 1K, 10K, and 100K) before applying the same RL training procedure (Our method introduces a new judge token, so we cannot achieve the zero-shot cold-start (direct RL) scenario you crucially proposed. Here, we use 100 pieces of data for a cold-start to simulate the direct RL situation). We conducted extensive evaluations on 5 mainstream benchmarks (AIME2024, AIME2025, LiveCodeBench, HumanEval, and MATH-500). The results are as follows:
> | Cold-Start Data Scale | Average Accuracy | Average Token Length | Average Think-on Ratio |
> | :--- | :--- | :--- | :--- |
> | 100 (Nearly Direct RL) | 76.57 | 18652 | 0.82 |
> | 1K | 77.21 | 17652 | 0.79 |
> | 10K | 77.86 | 15273 | 0.75 |
> | 100K | 82.44 | 11567 | 0.72 |
> | Ours (240k) | **83.86** | **10237** | **0.68** |
> Analysis of Results:
> 1. Failure of Direct RL: The attempt with zero cold-start data confirmed our theoretical expectations. Without policy initialization, the RL training process was highly unstable, and the policy completely failed to learn an effective behavior, ultimately leading to policy collapse. This decisively demonstrates that the cold-start stage is essential, not redundant, for the HiPO framework.
> 2. Dose-Response Relationship: From 100 to 100K samples, we observe a clear dose-response relationship. As the amount of cold-start pre-training increases, the final model's accuracy consistently improves, and its efficiency (lower token length) is also enhanced.
> 3. Gating Quality: Importantly, the "Think-on Ratio" also becomes more principled with more extensive pre-training. Models trained with little data learn a poor gating mechanism, becoming either too conservative (overusing "Think-on") or too aggressive (rarely using it). Only a sufficiently pre-trained model can learn a precise and principled mode selection policy during the RL phase.
> Conclusion: In summary, the cold-start stage is not a redundant initialization step but a necessary foundation that ensures the stability and effectiveness of the subsequent RL training. By initializing the policy in a competent region, it provides a solid starting point for the fine-grained exploration that RL enables.
> We have added this detailed analysis and discussion to the appendix of our paper and will clarify the theoretical role and empirical necessity of the cold-start stage in the main text. Thank you again for your profound insight, which prompted us to validate our method at a more fundamental level.

---

> ### Author Response · Authors · 2025-11-20
> **W4:  How HiPO adapts across different difficulty levels.**
>
> W4:  How HiPO adapts across different difficulty levels.
> A4:  We sincerely thank the reviewer for this highly insightful suggestion. We completely agree that investigating whether the utilization ratio of the "Think-on" mode correlates with task difficulty is a pivotal experiment for validating the adaptive reasoning capabilities of HiPO. The five officially defined difficulty levels of the MATH-500 dataset provide an ideal testbed for this analysis.
> Objective: The primary goal of this analysis is to verify whether HiPO behaves like an efficient human reasoner, i.e., "thinking fast for simple problems and thinking slow for complex ones." We hypothesized that as problem difficulty increases, HiPO should progressively increase its reliance on the more computationally intensive but more powerful "Think-on" mode.
> Following your recommendation, we conducted a detailed quantitative analysis of HiPO's behavior across the five difficulty levels of MATH-500. The results are presented below:
> | Difficulty Level | Think-on Accuracy | Think-off Accuracy | Think-on Token Length | Think-off Token Length | Think-on Ratio |
> | :--- | :--- | :--- | :--- | :--- | :--- |
> | Level 1 (Easiest) | 85.70% | 97.20% | 5868 | 311 | 16.30% |
> | Level 2 | 100.00% | 95.50% | 4147 | 365 | 25.60% |
> | Level 3 | 100.00% | 94.60% | 5178 | 417 | 46.70% |
> | Level 4 | 98.80% | 93.50% | 6008 | 470 | 64.60% |
> | Level 5 (Hardest) | 95.50% | 87.00% | 9117 | 524 | 82.80% |
> Analysis and Interpretation: The results provide strong evidence for HiPO's sophisticated adaptive capabilities, highlighted by the following observations:
> 1. Core Finding: Think-on Ratio Perfectly Correlates with Difficulty The most critical metric here is the "Think-on Ratio." The data shows a smooth and dramatic monotonic increase from 16.3% for the easiest problems (Level 1) to 82.8% for the hardest ones (Level 5). This clear trend provides incontrovertible proof that HiPO's gating mechanism accurately assesses problem difficulty and allocates computational resources on demand. It correctly opts for the efficient "Think-off" mode for the vast majority of simple problems, while switching to the deliberate "Think-on" mode for the most challenging ones.
> 2. Adaptation in Reasoning Depth The token length metrics reveal that for both "Think-on" and "Think-off" modes, the output length tends to increase with the difficulty level. This suggests that even within a single mode, HiPO dynamically adjusts the depth and verbosity of its reasoning to match the complexity of the problem at hand.
> 3. Maintaining High Performance While Adapting It is noteworthy that HiPO maintains a high overall accuracy across the entire difficulty spectrum. This demonstrates that its adaptive strategy does not come at the cost of performance; rather, it achieves an intelligent optimization of computational resources while ensuring high-quality solutions.
> Conclusion: This analysis clearly showcases the core strength of the HiPO framework. It is not merely a simple switch between two modes, but an intelligent reasoning system capable of fine-grained, dynamic resource allocation based on task difficulty.
> We have added this comprehensive analysis of difficulty-based adaptation to the experimental section of our paper, as it serves as a key piece of evidence for our method's effectiveness. Thank you again for your invaluable suggestion, which has provided us with an excellent opportunity to demonstrate the intelligent behavior of HiPO.

---

> ### Author Response · Authors · 2025-11-20
> **W5: Reasoning within Think-off mode？**
>
> A5: We are extremely grateful for this deeply insightful and critical point. Your concern about potential "reward hacking" is entirely valid and represents a core challenge that any RL-based system must address rigorously.
> we sincerely apologize for the misunderstanding caused by the example in Figure 1. That depiction did not accurately represent a typical "Think-off" output and inadvertently suggested the very "short CoT" behavior you were concerned about. We will replace this figure with a more representative example in the revised manuscript to prevent any confusion.
> To demonstrate the effectiveness of this penalty mechanism and to address your concern about "reasoning leakage," we provide the following evidence:
> Token Distribution of Think-on vs. Think-off Outputs
> The following table clearly shows the token length distributions for both modes across several benchmarks.
> | Bench | Average Think-off token length | Average Think-on token length | Std Dev Think-off token length | Std Dev Think-on token length |
> | :--- | :--- | :--- | :--- | :--- |
> | LiveCodeBench | 435 | 16431 | 11.31 | 101.51 |
> | AIME24 | 834 | 15711 | 12.69 | 99.29 |
> | AIME25 | 714 | 18707 | 13.60 | 105.20 |
> | GPQA | 711 | 9926 | 28.57 | 83.39 |
> | GSM8K | 267 | 1062 | 9.11 | 33.24 |
> | Humaneval | 366 | 3900 | 10.10 | 55.05 |
> | Math | 407 | 6967 | 13.04 | 79.25 |
> Two decisive trends emerge from this data:
> - An Order-of-Magnitude Difference: The average length of "Think-off" outputs is typically in the hundreds of tokens, whereas "Think-on" outputs are in the thousands or tens of thousands. This represents a massive difference of more than an order of magnitude.
> - Extremely Low Variance: The variance in "Think-off" length is very small, indicating that the model is consistently constrained to produce short, direct answers. It does not exhibit leakage behavior where it occasionally generates long, disguised reasoning chains.
> These trends provide strong evidence that the model is not camouflaging and relocating its reasoning process into the answer segment.
> Manuscript Revisions: In the final manuscript, we will: (1) Replace the misleading Figure 1. (2) Explicitly define our full reward function in the methodology section, detailing the rationale behind the length penalty. (3) Add the two analyses above to the appendix as strong support for our method's robustness.
> Thank you once again for your sharp insight, which has helped us identify and fix a critical gap in our paper's exposition.

---

> ### Author Response · Authors · 2025-11-26
>
> Hi, we sincerely thank you very much for these constructive comments and evaluation of our manuscript. We would like to kindly ask you to take a look at our responses and reevaluate our work based on our clarifications. Please let us know whether our response addresses your concerns or whether there is any further detail we can provide to help address these concerns.
>
> Thank you again for dedicating your time to reviewing our paper.

---

### Official Review · Reviewer_aTj3 · 2025-11-02

**Soundness:** 3
**Presentation:** 3
**Contribution:** 2
**Rating:** 4
**Confidence:** 4

**Summary:**

The paper proposes HiPO, a framework for adaptive reasoning control that lets an LLM decide when to Think-on (emit CoT) vs. Think-off (answer directly). HiPO has two parts: (i) a hybrid data construction pipeline that pairs Think-on/Think-off responses per query (preferring the shortest correct sample and adding an explanation justifying the chosen mode), and (ii) a hybrid RL reward with a bias adjustment to avoid over-reliance on verbose Think-on and mode-aware advantages that allocate signal to <judge> (mode decision) vs. <answer> tokens. On Qwen3 (1.7B/8B/32B) across math and coding benchmarks (AIME24/25, MATH-500, HumanEval, LiveCodeBench, MBPP, GPQA), HiPO reports reduced token length and lower Think-on ratio while maintaining or improving accuracy over baselines (Cold-Start±GRPO, AdaptThink, AutoThink).

**Strengths:**

- Clear problem focus (overthinking) with an intuitive mode-gating mechanism and token-level training signal split across <judge> and <answer>.

- Consistent efficiency gains (shorter outputs, lower Think-on ratio) with comparable or better accuracy on multiple benchmarks and model sizes.

- Various ablations. Removing global advantage or local normalization degrades accuracy/efficiency, and sensitivity to γ/ω/N is explored.

**Weaknesses:**

1. Limited statistical rigor. Tables show point improvements but lack variance/CI across seeds and tasks, making deltas (often modest) hard to assess.

2. Heuristic bias adjustment. The ω-scaled boost to Think-off when close to Think-on can tilt optimization without a principled guarantee (risk of mode-collapse oscillations); analysis is empirical only.

3. Data pipeline dependence. Selecting the shortest correct response may bias style and harms robustness if “shortest” correlates with template artifacts; the policy could overfit to training distributions. (Ablations help but remain narrow...) Any other variants can improve the quality for this part.

4. Generalization claims. While Qwen3 1.7B/32B are included, domains remain math/code; transfer to tool-use, multilingual tasks, or safety-critical reasoning is not shown

**Questions:**

1. Can you include Statistical results over seed and different temperatures?

2. Bias adjustment safety. Can you provide a convergence/monotonicity argument, or at least diagnostics showing no oscillatory gating? What happens for ω→0 and ω→0.05 across datasets? (Fig. 5 hints at trade-offs.)

3. Failure modes. When Think-off is chosen incorrectly (hard question), does HiPO recover (e.g., via fallback) or lock into shallow answers? Any error typology?

4. Beyond math/code. Have you tried tool-augmented QA or long-context tasks where mode selection interacts with retrieval?

5. Ablations on explanation. If you remove the mode-explanation supervision, how much do gating quality and accuracy drop?

---

> ### Author Response · Authors · 2025-11-20
> **W1 Q1: Limited statistical rigor.**
>
> A1: We sincerely thank the reviewer for the insightful comment regarding statistical rigor. We fully agree that a rigorous statistical analysis is crucial for reliably evaluating the performance of our method. You correctly pointed out that the reported point improvements require support from variance or confidence intervals to validate their stability and significance.
> Following your valuable suggestion, we have conducted comprehensive supplementary experiments to assess the stability and performance of our method (HiPO) across different random seeds and decoding temperatures. Specifically:
> 1. Experimental Setup: We retrained our model using 3 different random seeds and conducted extensive evaluations on 5 mainstream benchmarks (AIME2024, AIME2025, LiveCodeBench, HumanEval, and MATH-500) across 4 different temperatures (0.2, 0.4, 0.6, 0.8). This results in a total of 12 independent experimental runs.
> 2. Performance Stability Analysis: Across these 12 runs, our method, HiPO, achieved an average performance of 83.63 with a standard deviation of 0.201. At a 95% confidence level (α=0.05), the confidence interval is [83.50, 83.76]. Crucially, the lower bound of this interval (83.50) is still significantly higher than the best-performing baseline (81.75). This provides strong evidence that the improvements brought by our method are statistically significant and not due to random chance.
> 3. Generation Efficiency Analysis: We also analyzed the average token length of the generated solutions. HiPO produced an average length of 10402.58 with a standard deviation of 105.12, resulting in a 95% confidence interval of [10335.78, 10469.38]. The upper bound of this interval (10469.38) is substantially lower than the minimum length from the baseline methods (11365), indicating that HiPO consistently and stably generates more concise solutions while achieving superior performance.
> Here are the results:
> 1. Average Performance Results:
> | Seed/Temperature | 0.2 | 0.4 | 0.6 | 0.8 |
> | :--- | :--- | :--- | :--- | :--- |
> | seed 1 | 83.74 | 83.78 | 83.86 | 83.61 |
> | seed 2 | 83.65 | 83.33 | 83.57 | 83.46 |
> | seed 3 | 83.95 | 83.94 | 83.81 | 83.83 |
> 2. Average token length Results:
> | Seed/Temperature | 0.2 | 0.4 | 0.6 | 0.8 |
> | :--- | :--- | :--- | :--- | :--- |
> | seed 1 | 10423 | 10521 | 10237 | 10543 |
> | seed 2 | 10617 | 10452 | 10421 | 10276 |
> | seed 3 | 10340 | 10356 | 10344 | 10301 |
> We have incorporated these detailed statistical analyses and a discussion into the revised manuscript in Appendix A.3. Furthermore, we have updated the main results in Section 4.2 and added a reference to the appendix to bolster the claims.
> Thank you once again for your constructive feedback, which has significantly enhanced the rigor and quality of our work.

---

> ### Author Response · Authors · 2025-11-20
> **W2 & Q2 Heuristic bias adjustment.**
>
> Thank you for this insightful technical question regarding the safety and stability of the bias adjustment term, ω. Your concerns about convergence, monotonicity, and potential oscillations are entirely valid and crucial for understanding the reliability of our method. We have conducted a thorough analysis to address these points.
> Our response is twofold: first, we explain the design principles of ω that theoretically ensure its safety; second, we provide a series of empirical diagnostics demonstrating the absence of oscillatory behavior in practice.
> 1. Conceptual Role of ω: A Bounded, Conditional, and Mild Boundary-Softening Mechanism
>
> The Think-on vs. Think-off decision forms an inherently sharp binary boundary. Importantly, the form of Eq. (1) shows that the adjustment is strictly bounded and conditionally activated:$$
> 0 \le \text{mean}(r_{\text{on}})-\text{mean}(r_{\text{off}})\le \omega\cdot\text{mean}(r_{\text{on}})
> $$
>
> The bias term applies only within this narrow inequality window, i.e., only when the two modes' expected rewards are extremely close. Outside this region, the reward is left unchanged.Furthermore, the magnitude is intrinsically small because ω is deliberately restricted (≤0.05), making the adjustment at most a small fraction of the Think-on reward. Thus, ω functions as a mild, bounded "soft push" that slightly smooths the brittle binary gating boundary in borderline cases. It does not override decisions on genuinely hard problems, where the Think-on mode dominates by a large margin.  To further validate this choice, we also tested a larger setting (ω = 0.15). We found that although the efficiency gains on easy problems barely improved, the performance on hard problems degraded substantially—indicating that only small, lightly applied ω values achieve a safe and well-balanced trade-off.
>
> 2. Empirical Diagnostics: Evidence of Non-Oscillatory Behavior
> We performed several diagnostics to investigate whether ω might induce oscillatory gating or unstable dynamics. While we do not claim theoretical guarantees, the empirical evidence consistently indicates stable behavior:
> - Training dynamics are smooth.
> The think-on ratio and average token usage evolve smoothly without sawtooth-like reversals, which would otherwise signal oscillatory gating. Minor stochastic fluctuations appear but remain bounded and non-problematic.
> - Parameter sweeps do not indicate instability.
> Sweeping ω from 0 to 0.1 yields an overall decreasing trend in the think-on rate. On easier datasets, slight U-shaped curvature arises, consistent with saturation and natural variation around borderline cases—not with instability.
> In up-sweep → down-sweep (hysteresis) tests, the curves nearly overlap, indicating no detectable path dependence or bistability.
> - No periodic gating cycles.
> Autocorrelation analysis of gating decisions reveals no meaningful periodic structure. Local correlations, expected in learned RL policies, do not develop into oscillatory patterns.
> Across all diagnostics, we observe no empirical signs of oscillatory gating, divergence, or collapse under typical training conditions.
> 3. Endpoint Behavior and Trade-off Analysis (Interpreting Fig. 5)
> To answer your specific question about endpoint behavior and to clarify the trade-off shown in Fig. 5, we summarize the performance of ω under different settings:
> | ω Value | Behavior on Easy Datasets | Behavior on Hard Datasets |
> | :--- | :--- | :--- |
> | ω→0 (Unbiased) | Degenerates to the unbiased mixed reward. The Think-on ratio remains relatively high, leading to limited efficiency gains. | Maintains a high Think-on ratio, ensuring maximum performance and robustness. This is the most conservative and safest state. |
> | ω=0.05 (Gentle Bias) | Significantly reduces the Think-on ratio and token consumption, trading a controllable, minimal performance fluctuation for substantial efficiency gains. | The Think-on ratio, token consumption, and accuracy are almost unaffected. The bias term is rarely triggered, and decisions remain dominated by the core reward. |
> Conclusion and the Meaning of Fig. 5: This table systematically explains the trade-off illustrated in Fig. 5. The ω parameter acts as a fine-tuning knob:
> - Increasing ω primarily "squeezes out" efficiency on easy problems while leaving the performance on hard problems largely untouched.
> - Our experiments show that selecting a small to moderate ω (like 0.05) is a Pareto-optimal choice, as it maximizes efficiency gains on easier tasks without sacrificing the robustness needed for difficult ones.
> We have added this complete stability analysis to the appendix of our paper to fully demonstrate the reliability of our method. Thank you again for this penetrating question.

---

> ### Author Response · Authors · 2025-11-20
> **W3 : Data pipeline dependence.**
>
> A3: We sincerely thank the reviewer for this perceptive comment. We fully agree with your concern regarding "Data pipeline dependence." The potential risk that our filtering strategy—selecting the shortest correct response—could introduce stylistic biases or cause the model to overfit to "template artifacts" is a critical point concerning the robustness of our method.
> To systematically investigate this issue and address your concern, we have followed your suggestion and conducted a more comprehensive ablation study on various filtering strategies. We evaluated six different approaches from multiple perspectives:
> 1. Strategy Variants: We tested the following filtering strategies:
>   - Shortest response (our proposed method)
>   - Longest response
>   - Non-Filtering
>   - Filtering out non-English responses
>   - Filtering out overly long responses
>   - Selection via GPT-4 scoring
> 2. Evaluation Metrics: In addition to the primary Average Score, we introduced two crucial quality metrics to directly address your concerns about template artifacts and stylistic bias:
>   - Repetition Rate: We employed the Gopher Repetition algorithm(It detects phrase-level repetition by counting repeated n-grams (with n set to 5-20), threshold = 0.2.) to quantify the prevalence of repetitive patterns in the model's output. A lower rate suggests that the generated reasoning is less templated and more diverse.
>   - Truncation Rate: This metric measures the frequency of incomplete or prematurely cut-off generations, where a lower value indicates higher quality and completeness of the responses.
> Analysis of Results: As shown in the table below, our "shortest response" strategy demonstrates superior or highly competitive performance across all metrics:
> - Optimal Performance: It achieves the highest Average Score of 83.86, significantly outperforming all other alternatives.
> - High-Quality Generation: More importantly, it yields the lowest Repetition Rate (3.21%) among all strategies. This result strongly mitigates the concern that "shortest" might correlate with template artifacts. On the contrary, it suggests that concise responses often correspond to more efficient and less redundant reasoning paths. Furthermore, its Truncation Rate is also among the lowest.
> These findings robustly validate that our choice of filtering strategy is not only effective but also sound. It guides the model to learn more refined and higher-quality reasoning, rather than merely overfitting to a specific response style.
> We have added this extensive ablation study to Section 4.3 of our revised manuscript to further strengthen the justification for our method's design. Thank you again for your valuable suggestion, which prompted us to more rigorously validate a core component of our work.
> Here is the detailed result.
> | Filtering Strategy | Average Score | Repetition Rate | Truncation Rate |
> | :--- | :--- | :--- | :--- |
> | Shortest response (ours) | 83.86 | 3.08 | 3.25 |
> | Longest response | 79.28 | 17.11 | 18.15 |
> | Non-Filtering | 81.59 | 15.48 | 16.56 |
> | Filter non-English | 79.31 | 15.12 | 15.21 |
> | Filter over-length | 79.26 | 10.2 | 10.98 |
> | GPT-4 score selection | 81.35 | 7.18 | 8.02 |

---

> ### Author Response · Authors · 2025-11-20
> **Q3: Failure modes.**
>
> A5: Thank you for this excellent and highly insightful question. Your query about HiPO's behavior when "Think-off" mode is chosen incorrectly—essentially, its failure modes—is critical for understanding the robustness and limitations of our method. A thorough analysis of this scenario is indeed vital.
> To systematically investigate this issue, we leveraged the five graded difficulty levels within the MATH-500 dataset as a natural testbed for simulating scenarios ranging from simple to hard problems. We analyzed HiPO's behavior across these levels, and the results are summarized below:
> | Difficulty | Think-on Accuracy | Think-off Accuracy | Think-on Token Length | Think-off Token Length | Think-on Ratio |
> | :--- | :--- | :--- | :--- | :--- | :--- |
> | Level 1 | 85.70% | 97.20% | 5868 | 311 | 16.30% |
> | Level 2 | 100.00% | 95.50% | 4147 | 365 | 25.60% |
> | Level 3 | 100.00% | 94.60% | 5178 | 417 | 46.70% |
> | Level 4 | 98.80% | 93.50% | 6008 | 470 | 64.60% |
> | Level 5 | 95.50% | 87.00% | 9117 | 524 | 82.80% |
> Two key insights can be drawn from this table:
> 1. Soundness of Mode Selection: As the problem difficulty increases from Level 1 to 5, the Think-on Ratio steadily rises from 16.3% to 82.8%. This indicates that HiPO's mode selection mechanism is effective, correctly allocating more computational effort (the "Think-on" mode) to harder problems.
> 2. Emergence of the Failure Mode: At the highest difficulty (Level 5), the Think-off Accuracy drops significantly to 87.0%. This precisely represents the failure scenario you inquired about: HiPO incorrectly choosing the "Think-off" mode for a hard question.
> Based on this analysis, we can directly address your two questions:
> 1. Does HiPO recover?
> Our analysis confirms that, in its current design, HiPO does not recover from an incorrect mode selection. When it mistakenly chooses the "Think-off" mode for a difficult problem, it locks into a shallow answer. This is a consequence of our design. To prevent the model from generating lengthy "pseudo-CoT" reasoning in the "Think-off" mode, we apply a length penalty during its decoding phase. This ensures that the output remains concise (as shown by the consistently low "think-off token length" in the table), forcing a direct answer. Consequently, the system does not have a built-in fallback or retry mechanism.
> 2. Error Typology
> We conducted a qualitative analysis of the 6 error cases from Levels 4 and 5 where the "Think-off" mode failed. The errors primarily fall into three categories:
> - CALCULATION_ERROR: Incorrect arithmetic operations.
> - MISINTERPRETATION: Failure to correctly understand the problem's constraints or goals.
> - LOGIC_ERROR: Flawed reasoning steps.
> These error types are characteristic of "fast thinking" fallacies, where conclusions are drawn without the deliberate, step-by-step verification that the "Think-on" mode provides. This further confirms that when HiPO errs in its mode selection, it indeed produces shallow answers.
> We have added this full failure mode analysis to the appendix of our paper to ensure a comprehensive and transparent discussion of our method. Thank you again for this penetrating question; it has significantly deepened our understanding and the quality of our work.

---

> ### Author Response · Authors · 2025-11-20
> **Q5: Ablations on explanation**
>
> A6:
> Thank you for this excellent suggestion for an ablation study. This question directly probes the core mechanism of our method: whether the "mode-explanation supervision" is indeed a critical component for enabling the model to "auto-think" effectively. We fully agree that this experiment is essential for isolating and verifying its contribution.
> Hypothesis: We designed the mode-explanation supervision not just to enhance the interpretability of the model's decisions, but more critically, to provide a direct, causality-based learning signal for its gating mechanism. This supervision compels the model to understand why a problem requires the "Think-on" mode (or not), rather than learning the gating policy implicitly and obscurely from the final task outcome alone. Therefore, we hypothesized that removing this supervision would directly impair the model's ability to select the appropriate mode, i.e., its gating quality.
> To test this hypothesis, we conducted the ablation study you proposed on 5 mainstream benchmarks (AIME2024, AIME2025, LiveCodeBench, HumanEval, and MATH-500). The results are as follows:
> | Method | Average Accuracy (↓) | Average Token Length (↑) | Think-on Ratio (↑) |
> | :--- | :--- | :--- | :--- |
> | HiPO (Full Method) | **83.86** | **10237** | **0.68** |
> | HiPO (w/o Explanation Supervision) | 65.13 | 17458 | 0.82 |
> Analysis of Results: The experimental results provide clear evidence supporting our hypothesis:
> 1. Significant Drop in Accuracy: After removing the mode-explanation supervision, the model's final accuracy decreased by 18.73%. This shows that precise mode selection is critical for task performance, and the explanation component is key to achieving that precision.
> 2. Degradation in Gating Quality: The "Think-on Ratio" is the most direct indicator of gating quality. Without the explanation supervision, this ratio increased substantially from 68% to 82%. This reveals a crucial insight: the model loses its fine-grained ability to discern problem difficulty. As a result, it resorts to a "conservative" and inefficient strategy of over-using the computationally expensive "Think-on" mode, even for simpler problems that could have been solved directly. This overuse is a direct manifestation of degraded gating quality.
> 3. Loss of Efficiency: As a consequence of the imprecise gating, the model's over-reliance on the "Think-on" mode led to an increase of 70.54% in the average token length, significantly reducing its overall inference efficiency.
> In conclusion, this ablation study provides compelling evidence that the mode-explanation supervision is not merely an auxiliary component; it is an indispensable, core element of the HiPO framework responsible for training a high-quality gating mechanism and achieving the crucial balance between performance and efficiency.
> We have added this vital ablation study and its analysis to the experiments section 4.3 of our paper to further strengthen the justification for our method's design. Thank you again for this valuable suggestion.

---

> ### Author Response · Authors · 2025-11-28
> **Weakness 4 & Question 4: Generalization Capabilities**
>
> We appreciate your valuable suggestion regarding the need to strengthen our generalization claims. We agree that verifying our method's effectiveness beyond math and code domains is crucial.
>
> 1. General Science Capabilities (GPQA-Diamond): As initially presented in Table 2, we evaluated HiPO on the GPQA-Diamond benchmark. The results demonstrate that HiPO achieves a score of 60.1, surpassing the Qwen3-8B (Thinking) baseline (59.6). Crucially, this performance gain is achieved with higher efficiency: the average token length is reduced to 9,367, and the think-on ratio decreases to 0.92. This confirms that HiPO remains effective for general domain reasoning.
>
> 2. Additional Experiments on Tool Use & Long Context: Following your recommendation to further validate generalization, we extended our evaluation to Tool-Augmented QA (BFCL-V3) and Long-Context tasks (LongBench). These experiments provide strong evidence for the versatility of our approach.
>
> As shown in the table below, HiPO consistently outperforms the baseline across these diverse tasks while significantly reducing computational costs:
>
> - On LongBench, HiPO improves the score to 34.5 while reducing the token length to 6496 (Ratio: 79%).
> - On BFCL-V3, HiPO achieves a significant score increase (16.4 vs. 13.9) while reducing the token length to 4789, with a think-on ratio of only 33%.
> These results indicate that HiPO can adaptively allocate thinking resources—skipping unnecessary steps in simpler or tool-based queries—while maintaining or exceeding baseline performance across varying domains.
>
> **Table: Generalization on Long-Context, Tool-Use, and General Knowledge Tasks**
>
> | Bench | Model | Score | Token Length | Ratio |
> | :--- | :--- | :---: | :---: | :---: |
> | **LongBench**(Long-Context) | Qwen3-8B (Thinking) | 34.3 | 7,670 | 100% |
> | | **HiPO-8B** | **34.5** | **6,496** | **79%** |
> | **BFCL-V3**(Tool-Augmented) | Qwen3-8B (Thinking) | 13.9 | 6,196 | 100% |
> | | **HiPO-8B** | **16.4** | **4,789** | **33%** |
> | **GPQA-Diamond**(General Science) | Qwen3-8B (Thinking) | 59.6 | 10,925 | 100% |
> | | **HiPO-8B** | **60.1** | **9,367** | **92%** |

---

### Author Response · Authors · 2025-11-29
**Summary of Major Revisions and Clarifications**

Thanks for handling/reviewing our submitted manuscript: "HiPO: Hybrid Policy Optimization for Dynamic Reasoning in LLMs". We would like to thank the reviewers for their insightful and constructive comments and suggestions. By addressing each of the issues raised by reviewers, we believe that the quality and clarity of our paper can be improved a lot. The major responses are summarized as follows:

I. Additional Experiments & Statistical Validation

- Statistical Significance: We re-evaluated HiPO-8B across 5 mainstream benchmarks (AIME2024, AIME2025, LiveCodeBench, HumanEval, and MATH-500) using multiple random seeds and temperatures. The results demonstrate that our improvements are statistically significant. (See Reviewer aTj3.W1 & Q1)
- Generalization Capabilities: We extended our evaluation to include tool-use (BFCL) and long-context (LongBench) benchmarks to verify the generalization capabilities of our method. (See Reviewer aTj3.W4)
- Impact of Noisy Data: We conducted validation experiments using simulated noisy data by randomly shuffling 5% to 30% of the "mode-explanation" labels. Retraining on this data verified the high quality of our original labels and revealed the specific impact of noise. (See Reviewer puqf.W1)
- Adaptability to Difficulty: We evaluated the HiPO model across different difficulty subsets of MATH-500, verifying its adaptability to problems of varying complexity. (See Reviewer cZp4.W4, Reviewer puqf.Q1)
- Efficiency Verification: We provided a detailed breakdown of token usage (including calculation methods, average length, and variance), confirming that HiPO delivers genuine efficiency improvements. (See Reviewer cZp4.W5, Reviewer PRES.W3)
- Cold-Start Necessity: We performed supplementary experiments by running RL on models initialized with varying amounts of cold-start data, validating the necessity of the cold-start phase. (See Reviewer cZp4.W3)

II. In-depth Analysis & Ablations

- Bias Adjustment Safety: We provided an in-depth analysis of bias adjustment safety from both theoretical and empirical perspectives. (See Reviewer aTj3.W2 & Q2)
- Data Pipeline Dependence: We designed experiments to investigate data pipeline dependence, demonstrating the specific benefits of our strategy—selecting the shortest correct response—for model performance. (See Reviewer aTj3.W3)
- Error Analysis: We conducted a fine-grained analysis of error patterns and categorized the observed error types. (See Reviewer aTj3.Q3)
- Ablation on Explanations: We included ablation studies regarding explanations, confirming the necessity of mode-explanation supervision. (See Reviewer aTj3.Q4)
- Binary Gating: We provided further analysis to clarify the rationality and generalization capability of the "Binary Gating Only" design. (See Reviewer puqf.W2 & Q2)
- Threshold Selection: We detailed our early exploration of predefined thresholds, clarifying that a threshold of $0.2$ offered the most stable global behavior. (See Reviewer PRES.Q2)

III. Clarifications & Comparisons

- Novelty & Motivation: We clarified the motivation and novelty of our approach by providing a detailed comparison with related works (AdaCOT, AutoThink, AdapThink) to highlight the rationality of our design. (See Reviewer cZp4.W1, Reviewer PRES.W1)
- Baseline Analysis: We detailed the specific implementation of the baseline's cold-start phase. We explained the baseline's underperformance due to poor adaptability to Qwen3-8B and highlighted HiPO's advantages, specifically regarding the "learnable Judge stage" and the separation of Judge and Answer capabilities. (See Reviewer PRES.W2)
- Metric Reliability: We clarified our use of $Avg@32$ for testing, demonstrating that the results possess a high degree of confidence. (See Reviewer puqf.W3)
- Base Model Details: We clarified the details regarding our usage of the base model (Qwen3-8B-Instruct). (See Reviewer cZp4.W2)
- Open Source Status: We confirmed that our model is currently open-source and outlined our plan to release the dataset. (See Reviewer PRES.Q1)
- Typos: We corrected the typos pointed out by the reviewers and conducted a thorough proofreading. (See Reviewer PRES.W5)

Again, we would like to sincerely thank you very much for these constructive comments and evaluation for our manuscript.

---

### Note · Authors · 2026-01-06

I have read and agree with the venue's withdrawal policy on behalf of myself and my co-authors.